# Twice-Sequential Monte Carlo for Tree Search

**Yaniv Oren** [1]  **Joery A. de Vries** [1]  **Pascal R. van der Vaart** [1]  **Matthijs T. J. Spaan** [1]  **Wendelin Böhmer** [1]

## Abstract

Model-based reinforcement learning (RL) methods that leverage search are responsible for many milestone breakthroughs in RL. Sequential Monte Carlo (SMC) recently emerged as an alternative to the Monte Carlo Tree Search (MCTS) algorithm which drove these breakthroughs. SMC is easier to parallelize and more suitable to GPU acceleration. However, it also suffers from large variance and path degeneracy which prevent it from scaling well with increased search depth, i.e., increased sequential compute. To address these problems, we introduce Twice Sequential Monte Carlo Tree Search (TSMCTS). Across discrete and continuous environments TSMCTS outperforms the SMC baseline as well as a popular modern version of MCTS as a policy improvement operator, scales favorably with sequential compute, reduces estimator variance and mitigates the effects of path degeneracy while retaining the properties that make SMC natural to parallelize.

## 1. Introduction

The objective of Reinforcement Learning (RL) is to approximate optimal policies for decision problems formulated as interactive environments. For this purpose, model-based RL algorithms that use *search* (also called *planning*) with a model of the environment's dynamics for policy improvement have been tremendously successful. Examples include games (Silver et al., 2016), robotics (Hubert et al., 2021) and algorithm discovery (Fawzi et al., 2022; Mankowitz et al., 2023). These milestone approaches are all based in the Alpha/MuZero (A/MZ, Silver et al., 2018; Schrittwieser et al., 2020) algorithm family and are driven by Monte Carlo

---

An open-source implementation of the algorithms presented in this paper is publicly available at https://github.com/joeryjoery/tsmcts.

[1]Delft University of Technology, Delft, The Netherlands. Correspondence to: Yaniv Oren <y.oren@tudelft.nl>.

*Proceedings of the $43^{rd}$ International Conference on Machine Learning*, Seoul, South Korea. PMLR 306, 2026. Copyright 2026 by the author(s).

Tree Search (MCTS, see Świechowski et al., 2023). Like many search algorithms, the main bottleneck of MCTS is intensive compute and therefore runtime cost. Due to the sequential nature of MCTS (Liu et al., 2020; Macfarlane et al., 2024), it is challenging to address its runtime cost through parallelization and GPU acceleration (for example, with JAX (Bradbury et al., 2018)) which are staples of modern deep RL. In addition, MCTS requires maintaining the entire search tree in memory. GPU-acceleration approaches often require static shapes for best performance which forces memory usage to scale with the tree size and makes space complexity another bottleneck for GPU scalability.

To address this, alternative search algorithms for policy improvement have emerged (Piché et al., 2019; Macfarlane et al., 2024). These algorithms use Sequential Monte Carlo (SMC, see Chopin & Papaspiliopoulos, 2020) for policy optimization in the Control as Inference (CAI, see Levine, 2018) probabilistic inference framework for RL. SMC is used to approximate a distribution over trajectories generated by an improved policy at the root using $N$ particles in parallel. The parallel nature and lower memory cost, which scales linearly with $N$, make SMC well suited for parallelization and GPU acceleration (Macfarlane et al., 2024) and competitive with MCTS for policy improvement (Macfarlane et al., 2024; de Vries et al., 2025).

SMC however suffers from two major problems: *sharply increasing variance with search depth* and *path degeneracy* (Chopin & Papaspiliopoulos, 2020). The variance increase stems from the exponential growth in the number of possible trajectories $s_{1:T}$ in the search depth $T$. Path degeneracy is a phenomenon where eventually all particles become associated with the same state-action at the root of the search tree. This renders any additional search obsolete and collapses the root policy into a delta distribution causing target degeneracy (de Vries et al., 2025). These problems can cause the performance of SMC to *deteriorate* rather than *scale* with sequential compute (search depth). In contrast, MCTS scales well with sequential compute and does not suffer from path degeneracy.

To address these limitations we design a novel SMC-based search algorithm for policy improvement in RL: *Twice Sequential Monte Carlo Tree Search* (TSMCTS). We begin with a reformulation of SMC for search in RL which de-

necessitates the framework of CAI. The new formulation is simpler, optimizes for the RL objective directly and allows for general policy improvement operators expanding the connection between SMC and policy improvement. This new formulation also facilitates a shift in the perspective of the search from estimating *trajectories* to estimating *action values* of an improved policy at the root. The new perspective in turn facilitates incorporating a backpropagation mechanism akin to that of MCTS for value aggregation at the root, which mitigates both variance and path degeneracy. We call this intermediate algorithm *SMC Tree Search* (SMCTS). Next, inspired by the insight that given known search budget, action selection at the root can be viewed as a *simple-regret*, *best-action-identification* problem (Danihelka et al., 2022) the final algorithm TSMCTS utilizes the Sequential Halving (Karnin et al., 2013) bandit algorithm to improve policy improvement at the root. TSMCTS sequentially calls SMCTS at the root on a halving number of actions with a doubling number of particles, independently in parallel, which also addresses the remaining effects of path degeneracy at the root and further reduces variance.

We evaluate TSMCTS on a popular environment suite containing a range of continuous and discrete environments. TSMCTS significantly outperforms popular, modern search-based baselines such as SPO (Macfarlane et al., 2024), GumbelAZ (Danihelka et al., 2022) and TRT SMC (de Vries et al., 2025), outperforming the respective search-based operators for policy improvement: the SMC baseline used by SPO, GumbelMCTS used by GumbelAZ and TRT SMC's search, in an equal footing comparison. TSMCTS scales well with additional sequential compute, unlike the SMC baseline which deteriorates, while maintaining the same space and runtime complexity properties that make SMC well suited for parallelization. TSMCTS demonstrates significantly reduced estimator variance and mitigates path degeneracy, resulting in an overall significantly improved search algorithm for policy improvement in RL.

## 2. Background

In RL, the environment is represented by a Markov Decision Process (MDP, Bellman, 1957) $\mathcal{M} = \langle \mathcal{S}, \mathcal{A}, \rho, R, P, \gamma \rangle$. $\mathcal{S}$ is a set of states, $\mathcal{A}$ a set of actions, $\rho$ an initial state distribution, $R : \mathcal{S} \times \mathcal{A} \to \mathbb{R}$ a bounded possibly stochastic reward function, and $P$ is a transition distribution such that $P(s'|s, a)$ specifies the probability of transitioning from state $s$ to state $s'$ with action $a$. The policy of the agent $\pi \in \Pi$ is defined as a distribution over actions $a \sim \pi(s)$. Its optimality is defined with respect to the objective $J_\pi$, the maximization of the *expected discounted return* (also called value $V^\pi$) over the initial state distribution $\rho$:

$$J_\pi = \mathbb{E}[V^\pi(s_1)|s_1 \sim \rho] = \mathbb{E}_{\pi, P, \rho}\Big[\sum_{t=1}^{H} \gamma^t R(s_t, a_t)\Big], \quad (1)$$

where $\mathbb{E}_{\pi, P, \rho}$ denotes the expectation with respect to $a_t \sim \pi(s_t), s_{t+1} \sim P(s_t, a_t)$ and $s_1 \sim \rho$. The discount factor $0 < \gamma < 1$ is used in infinite-horizon MDPs, i.e. $H \to \infty$, to guarantee that the values remain bounded, and otherwise $\gamma = 1$. A state-action *Q-value function* is defined as follows: $Q^\pi(s, a) = \mathbb{E}_P[R(s, a) + \gamma V^\pi(s')]$. We denote the value of the optimal policy $\pi^*$ with $V^*(s) = \max_\pi V^\pi(s)$.

**Policy improvement and Greedification** Policy improvement is used to motivate the convergence of RL algorithms, abstracted as *approximate policy iteration* algorithms, to the optimal policy (see Danihelka et al., 2022; Oren et al., 2025b). *Policy improvement operators* are often defined as operators $\mathcal{I} : \Pi \times \mathcal{Q} \to \Pi$ such that $\forall s \in \mathcal{S} : V^{\mathcal{I}(\pi, Q^\pi)}(s) \geq V^\pi(s)$ and $\exists s \in \mathcal{S} : V^{\mathcal{I}(\pi, Q^\pi)}(s) > V^\pi(s)$, unless $\pi$ is already an optimal policy. We define $\mathcal{Q}$ generally as the set of all bounded functions on the state-action space $q \in \mathcal{Q} : \mathcal{S} \times \mathcal{A} \to \mathbb{R}$, to indicate that policy improvement operators are defined for exact $Q^\pi$ as well as approximate $q \approx Q^\pi$.

The *policy improvement theorem* (Sutton & Barto, 2018) proves that a tractable class of operators, *greedification operators* (Chan et al., 2022; Oren et al., 2025b) are policy improvement operators. Greedification operators $\mathcal{I}$ are operators over the same space, such that the policy $\pi'(s) = \mathcal{I}(\pi, q)(s)$ is *greedier* than $\pi(s)$ with respect to $q(s, a)$, such that:

$$\forall s \in \mathcal{S} : \sum_{a \in A} \pi'(a|s)q(s, a) \geq \sum_{a \in A} \pi(a|s)q(s, a), \quad (2)$$

$$\exists s \in \mathcal{S} : \sum_{a \in A} \pi'(a|s)q(s, a) > \sum_{a \in A} \pi(a|s)q(s, a), \quad (3)$$

unless $\pi$ is already a greedy ($\arg\max$) policy with respect to $q$. Policy improvement is guaranteed when $q = Q^\pi$. We define *strict* greedification operators $\mathcal{I}$ as operators that satisfy a strict $>$ Inequality 2, unless $\pi$ is already a greedy policy at $s$.

A popular strict greedification operator which trades off between *greedification* (maximizing $\sum_{a \in \mathcal{A}} \pi'(a|s)q(s, a)$) and regularization with respect to $\pi$ is the *regularized policy improvement* operator (Grill et al., 2020):

$$\mathcal{I}_{GMZ}(\pi, q)(a|s) \propto \pi(a|s) \exp\big(\beta q(s, a)\big). \quad (4)$$

We will use greedification operators to drive the policy improvement produced by SMC and TSMCTS, and prove that these algorithms are themselves policy improvement operators, albeit not necessarily greedification opeartors.

**Search for policy improvement** RL algorithms which have access to a dynamics model of the environment $\mathcal{M} = (P, R)$ (learned, latent, exact and / or given) often use *search* for policy improvement (Moerland et al., 2023). Search

refers to the process of look-ahead in the model from a state $s$, often using a prior policy $\pi_\theta$ and value DNN $v_\phi$, in order to extract an improved policy $\pi'(s)$ at the current state $s$. A popular and extremely successful example is the Alpha/MuZero (A/MZ) line of work, which use the Monte Carlo Tree Search (MCTS) algorithm for policy improvement.

MCTS maintains a search tree over states with its root $s$, the current state in the environment. The tree is constructed iteratively by (i) *searching* the existing tree using an improved policy (Grill et al., 2020). (ii) *Expanding* an as-of-yet unobserved transition $(s_{T-1}, a_{T-1} s_T)$ which gathers new information from the model, in the form of reward $r(s_{T-1}, a_{T-1})$ and value $V^{\pi_\theta}(s_T)$. In modern variants, the evaluation of unexpanded nodes $s_T$ is done using $v_\phi(s_T) \approx V^{\pi_\theta}(s_T)$. (iii) The bootstrapped return from this search trajectory $\nu(s_1, a_1) = \sum_{t=1}^{T} \gamma^{t-1} r(s_t, a_t) + \gamma^T v_\phi(s_T)$, where $s_1 := s$, induced by the improved policy, is *backpropagated* through all nodes along the trajectory $s_1, \ldots, s_T$. The nodes maintain an estimator of the states and action values in the form of the average of all returns $\nu^i(s_t)$ observed during search.

Search algorithms generally return an action $a$ to take in the environment, an improved policy $\pi_{search}(s)$ and the root value $V_{search}(s)$. $\pi_{search}(s)$ is used to train the prior policy $\pi_\theta$ (as targets in a cross entropy loss) and $V_{search}(s)$ is used to produce bootstraps for TD-targets (Schrittwieser et al., 2020) or even value targets directly (Oren et al., 2025a). Modern MCTS variants (Danihelka et al., 2022) use $\pi_{search}(s) = \mathcal{I}_{GMZ}(\pi_\theta, q)(s)$ and $V_{search}(s) = \sum_{a \in A} \pi_{search}(a|s)q(s, a)$, where $q(s, a)$ is the average of all bootstrapped-returns through $(s, a)$.

**Sequential Halving for policy improvement at the root**
The search policy of MCTS was originally designed to optimize for *cumulative* regret. Danihelka et al. (2022) observed that since the search budget is often known in advance however, it is better to optimize the search at the root for *simple regret* rather than cumulative regret. Motivated by that observation, they developed GumbelMCTS, respectively driving the learning of a new variant of A/MZ (GumbelA/MZ).

GumbelMCTS replaces the search policy at the root with the Sequential-Halving (SH, Karnin et al., 2013) *simple-regret* minimization algorithm which provides almost-optimality guarantees for simple-regret best-action identification in a bandit with stationary returns. Once search has concluded, GumbelMCTS returns the action identified by SH to take in the environment as well as the improved policy $\pi_{search}(s)$ and root value $V_{search}(s)$ as defined above.

**Sequential Monte Carlo** (SMC) methods approximate a sequence of *target distributions* $p_t(x_{1:t})$ using *proposal distributions* $u_t(x_{1:t})$. At each time step $t \in \{1, \ldots, T\}$, $N$ particles $x_t^n$ with weights $w_t^n$ are updated via *mutation, correction, and selection* (Chopin, 2004). *Mutation*: each trajectory $x_{1:t-1}^n$ is extended by sampling $x_t^n \sim u_t(x_t \mid x_{1:t-1}^n)$. *Correction*: The weights are updated using *importance sampling* such that the set of weighted particles $\{x_t^n, w_t^n\}_{n=1}^N$ approximates expectations under the target:

$$w_t^n = w_{t-1}^n \cdot \frac{p_t(x_t^n \mid x_{1:t-1}^n)}{u_t(x_t^n \mid x_{1:t-1}^n)}, \tag{5}$$

$$\frac{\sum_{n=1}^N w_t^n f(x_t^n)}{\sum_{n=1}^N w_t^n} \approx \mathbb{E}_{p_t}[f(x_t)], \tag{6}$$

where $f(x_t)$ is any function of interest. *Selection*: The particles are resampled proportionally to the normalized weights: $\{x_t\}_{n=1}^N \sim$ Multinomial($N$, normalized $w_t$), $\{w_t^n = 1\}_{n=1}^N$ to prevent weight impoverishment. We refer to (Chopin & Papaspiliopoulos, 2020) for more details.

**SMC as a search algorithm for policy improvement**
Piché et al. (2019) adapted SMC as a search algorithm for policy improvement at state $s$ by defining the target $p_t(\tau_t)$ and proposal $u_t(\tau_t)$ as distributions over trajectories $\tau_t = (s_1, a_1, \ldots, s_t, a_t) = x_{1:t}$ starting in a given state $s_1 = s$ (the "root" state from the environment). Piché et al. (2019) derived the target $p_t(\tau_t)$ using the framework of Control As Inference (CAI, see (Levine, 2018)), which induces the following weight update:

$$w_t^n \propto w_{t-1}^n \frac{\pi(a_t^n \mid s_t^n)}{\pi_\theta(a_t^n \mid s_t^n)} \mathbb{E}_{s_{t+1}^n} \big[ \exp(A(s_t^n, a_t^n, s_{t+1}^n)) \big].$$

$\pi$ is a reference policy required by CAI. In the maximum entropy setup, $\pi$ is the uniform policy, which recovers the maximum entropy solution (Haarnoja et al., 2018). $A$ is the soft-advantage using soft values (see (Levine, 2018)):

$$A(s_t, a_t, s_{t+1}) = r_t + V(s_{t+1}) - \log \mathbb{E}_{s_t}[\exp(V(s_t))].$$

See (Piché et al., 2019) for more detail and full derivation. We refer to this algorithm as **CAI-SMC** to distinguish it from the SMC-based algorithms developed in this work.

CAI-SMC produces a policy $\hat{\pi}_{SMC}^T$ at the root $s = s_1$ after $T$ steps as a weighted mixture of point-masses:

$$\hat{\pi}_{SMC}^T(a|s) := \sum_{n=1}^N \overline{w}_T^n \mathbb{1}_{\tau_T^n(a_1)=a} \approx \mathbb{E}_{p_T(\tau_T)}[w_T] \tag{7}$$

$$\approx \pi(a|s)\mathbb{E}_{s'} \big[ \exp(A(s, a, s')) \big], \tag{8}$$

where $\tau_T(a_1)$ denotes the first action in the trajectory and $\bar{w}_t$ are the normalized particle weights and $\pi'$ is the *soft-optimal policy* with respect to $\pi$ (Levine, 2018).

Piché et al. (2019) used $\hat{\pi}_{SMC}^T(s)$ to select actions in the environment, Soft Actor Critic's (Haarnoja et al., 2018) policy

gradient to train $\pi_\theta$ and environment interactions to train the model $(\hat{P}, \hat{R}) \approx (P, R)$. Macfarlane et al. (2024) showed that with $\pi = \pi_\theta$ CAI-SMC can be used as a policy improvement operator to generate targets for $\pi_\theta$ in an Expectation Maximization (Dempster et al., 1977) loop in their method SPO, similar to the manner in which MCTS is used by AZ.

# 3. Sequential Monte Carlo Search for Generalized Policy Improvement

We begin by extending prior work's formulation of SMC as a search algorithm for RL (Piché et al., 2019; Macfarlane et al., 2024) beyond the framework of CAI. This formulation is simpler, optimizes for $J_\pi$ the optimization objective of RL directly and accepts general greedification operators $\mathcal{I}$. In addition, it facilitates a perspective shift from *reasoning over a distribution over trajectories* to *reasoning over the values of actions* from an improved policy, which we will build on in the following sections.

We formulate the proposal $u_t(\tau_t)$ and target $p_t(\tau_t)$ similarly as distributions over trajectories $\tau_t$. We define $u_t(\tau_t)$ in the same manner as prior work:

$$u_t(\tau_t) = \pi_\theta(a_1|s_1)\Pi_{i=1}^{t-1}P(s_{i+1}|s_i,a_i)\pi_\theta(a_{i+1}|s_{i+1}),$$
$$u_t(\tau_t|\tau_{t-1}) = P(s_t|s_{t-1},a_{t-1})\pi_\theta(a_t|s_t). \quad (9)$$

The choice of prior work to use CAI to derive the weight update induces a specific target distribution $p_t(\tau_t)$. In contrast, we define here the target distribution more generally as a distribution induced by an improved policy $\pi'(s) = \mathcal{I}(\pi_\theta, Q^{\pi_\theta})(s)$ using any greedification operator $\mathcal{I}$:

$$p_t(\tau_t) = \pi'(a_1|s_1)\Pi_{i=1}^{t-1}P(s_{i+1}|s_i,a_i)\pi'(a_{i+1}|s_{i+1}),$$
$$p_t(\tau_t|\tau_{t-1}) = P(s_t|s_{t-1},a_{t-1})\pi'(a_t|s_t). \quad (10)$$

Given $p_t(\tau_t|\tau_{t-1})$ and $u_t(\tau_t|\tau_{t-1})$, the importance sampling weights $w_t^n$ of SMC derive as follows:

$$w_t^n = w_{t-1}^n \frac{p_t(\tau_t^n|\tau_{t-1}^n)}{u_t(\tau_t^n|\tau_{t-1}^n)} \quad (11)$$

$$= w_{t-1}^n \frac{P(s_t^n|s_{t-1}^n,a_{t-1}^n)\pi'(a_t^n|s_t^n)}{P(s_t^n|s_{t-1}^n,a_{t-1}^n)\pi_\theta(a_t^n|s_t^n)} \quad (12)$$

$$= w_{t-1}^n \frac{\pi'(a_t^n|s_t^n)}{\pi_\theta(a_t^n|s_t^n)}. \quad (13)$$

In practice the improved policy $\pi'$ is approximated with DNNs $Q^\pi(s,a) \approx q_\phi(s,a)$ or $r(s,a) + \gamma v_\phi(s')$ as in CAI-SMC and A/MZ. Since this derivation does not rely on CAI, the values are the regular RL values defined in Equation 1. We refer to this algorithm as **RL-SMC** (Algorithm 1). Equation 13 reduces to CAI-SMC's weight update for the soft-advantage operator (see Appendix A.2 for derivation).

**Policy improvement**  RL-SMC returns the improved policy at the root-state $s$:

$$\hat{\pi}_{SMC}^T(a|s) := \sum_{n=1}^N \overline{w}_T^n \mathbb{1}_{\tau_T^n(a_1)=a} \approx \mathbb{E}_{p_T(\tau_T)}[w_T] \quad (14)$$
$$=: \pi_{SMC}^T(a|s) \propto \pi_\theta(a|s)\exp(Q^{\pi'}(s,a)),$$

in the same manner as CAI-SMC (Equation 7), where $Q^{\pi'}$ is the value of the policy defined as $\pi'$ at all states $s_{1,...,T}$ (the planning horizon) and $\pi_\theta$ at all states $s_{T+1,...}$. We proceed to establish that for any greedification operator $\mathcal{I}$, the target policy approximated by RL-SMC is an improved policy:

**Theorem 1.** *For any greedification operator $\mathcal{I}$, horizon $T$, prior policy $\pi_\theta$, true dynamics model $(P, R)$ and evaluation $Q^{\pi_\theta}$ the target policy $\pi_{SMC}^T(s)$ approximated by RL-SMC at the root $s$ is an improved policy.*

**Intuition**  RL-SMC produces a distribution over trajectories $p_T(\tau_T)$ from a policy $\pi'$ that is improved with respect to the prior policy $\pi_\theta$ at all states $\{s_1 = s, s_2, \ldots, s_T\}$. Since this policy is improved with respect to the future $\{\ldots, s_T\}$, it is of course also improved at the current state in the environment $s$. See Appendix A.1 for a complete proof.

The proof of Theorem 1 points to one of the advantages of using search for policy improvement. By unrolling with the model, RL-SMC produces a policy that is improved for $T$ *consecutive time steps*:

**Corollary 1.** *For any strict greedification operator $\mathcal{I}$, horizon $T$, prior policy $\pi_\theta$, true dynamics model $(P, R)$ and evaluation $Q^{\pi_\theta}$ the policy approximated by RL-SMC is T-steps improved:*

$$V^{\pi_{SMC}^T}(s) > \cdots > V^{\pi_{SMC}^1}(s) > V^{\pi_\theta}(s), \quad (15)$$

*as long as $\pi_\theta$ is not already an $\arg\max$ policy with respect to $Q^{\pi_\theta}$ at all states $s_1, \ldots, s_T$.*

The proof follows directly from applying strict greedification operators (strict Inequality 2) in the proof of Theorem 1.

**Large variance and path degeneracy**  The estimator $\hat{\pi}_{SMC}^T(s)$ produced by SMC suffers however from two major problems: *large variance* in $T$ and *path degeneracy*.

SMC methods are prone to high variance, increasing up to exponentially as a function of $t$, a manifestation of the curse of dimensionality and the exponentially growing size of the domain $\tau_t$ (Chopin & Papaspiliopoulos, 2020).

SMC is also prone to *path degeneracy* (Chopin & Papaspiliopoulos, 2020): consecutive selection steps will eventually concentrate all particles to trajectories that are associated with one root action $a_1^i = a^i$. Once all particles are associated with the same root action $a^i$, say at step $h$,

the estimator $\hat{\pi}^h_{SMC}(a^i|s) = 1$ and zero for all other root actions $a \in \mathcal{A}, a \neq a^i$. From that point on, the estimator $\hat{\pi}^t_{SMC}(a|s)$ will not change for all additional search steps $t > h$ because there are no trajectories remaining starting in actions other than $a^i$. Thus particles can never be re-sampled out of trajectories starting in this action. This is problematic for two reasons: (i) As the search has no effect from $t > h$. That is, the algorithm cannot benefit from additional sequential compute in the form of search depth $T > h$. (ii) It results in a delta distribution policy target that is a crude approximation for any underlying improved policy $\pi_{search}(s)$ but an $\arg\max$, which can directly cause training-target degeneracy, see (de Vries et al., 2025).

In the following sections, we design an algorithm which addresses both problems.

## 4. Value-Based Sequential Monte Carlo

The source of both the large variance and the effects of path degeneracy are due to SMC optimizing a distribution over trajectories. We can mitigate both problems by switching the perspective of the search from optimizing *trajectories* to optimizing *values*: (i) The value $Q^{\pi^t_{SMC}}$ does not stop updating when all particles are associated with one action at $t = h$ and thus search for $t > h$ is not obsolete. This allows SMC to benefit from increased search depth. (ii) Information is not lost about actions that have no remaining particles which prevents target degeneracy. This is similar to the idea recently proposed by de Vries et al. (2025) in the guise of policy log-probabilities in the framework of CAI. (iii) By aggregating values during search the variance in the prediction at the root can be reduced.

The particles can be used to approximate the value $Q^{\pi^t_{SMC}}(s_1, a)$ at the root $s_1 := s$ as follows:

$$\forall a \in A_1 : \quad Q^{\pi^t_{SMC}}(s_1, a) = \tag{16}$$

$$= \mathbb{E}_{\pi^t_{SMC}, P}\Big[ \sum_{i=1}^{t} \gamma^i r_i + \gamma^{t+1} V^{\pi_\theta}(s_{t+1}) \,\big|\, s_1, a \Big] \tag{17}$$

$$\approx \sum_{n=1}^{N} \bar{w}^n_t \mathbb{1}_{\tau^n_t(a_1)=a} \sum_{i=1}^{t} \gamma^i r^n_i + \gamma^{t+1} V^{\pi_\theta}(s^n_{t+1}) \tag{18}$$

$$:= Q_t(s_1, a), \tag{19}$$

where $A_1$ are the actions sampled by the particles at the first step and $\bar{w}_t$ are the weights normalized. To reduce variance (see Appendix A.6 for more detail), instead of $Q_t$ we can keep track of the *average* value observed during search $\bar{Q}_t$:

$$\bar{Q}_t(s_1, a) = \frac{1}{t} \sum_{i=1}^{t} Q_i(s_1, a). \tag{20}$$

Whenever there are no particles associated with action $a$,

the value $\bar{Q}_t(s_1, a)$ is not updated. The average value can be updated in place, maintaining SMC's space complexity.

This value-based extension to RL-SMC can be thought of as iterating: (i) *Expansion*: Transition to $s_t$ and sample from the prior-policy $\pi_\theta(s_t)$. (ii) *Search*: compute importance sampling weights to align with the improved policy $\pi'(s_t)$. (iii) *Backpropagation*: evaluate the returns at states $s_t$, average the return across the particles associated with the same action $a$ at the root and incorporate it into the running mean $\bar{Q}_t$. Due to the similarity between this three-step process and MCTS', we refer to this algorithm as Sequential-Monte-Carlo Tree Search (**SMCTS**, summarized in Algorithm 2).

**Policy improvement** $\bar{Q}_t$ estimates the value of a mixture of increasingly improving policies $\pi^T_{SMCTS}(s) = \frac{1}{N} \sum_{i=1}^{T} \pi^i_{SMC}(s)$ which is itself an improved policy (see Lemma 1 in Appendix A.3). SMCTS returns an improved policy at root state $s$ using greedification with respect to $\bar{Q}_t$:

$$\pi_{search}(s) = \mathcal{I}(\pi_\theta, \bar{Q}_T)(s), \tag{21}$$

$$V_{search}(s) = \sum_{a \in A_1} \bar{Q}_T(s, a) \pi_{search}(a|s). \tag{22}$$

Extracting policy improvement with respect to policy $\pi_\theta$ and the value of an improved policy is a standard choice in model-based RL (Silver et al., 2016; Danihelka et al., 2022; de Vries et al., 2025). We establish that this choice is sound:

**Theorem 2.** *Given a policy $\pi(s)$, the value of an improved policy $Q^{\pi'}(s, a)$ and a greedification operator $\mathcal{I}$ the policy $\pi''(s) = \mathcal{I}(\pi, Q^{\pi'})(s)$ is an improved policy.*

See Appendix A.3 for proof.

## 5. Twice-Sequential Monte Carlo Tree Search

Fundamentally, the objective of the search is policy improvement *at the root $s$*. In contrast, SMC approximates trajectories from the $T$-steps improved policy $\pi^T_{SMC}$ and SMCTS approximates the value of the improved mixed policy $\pi^T_{SMCTS}$. To better spend the search resources for optimization *at the root*, we leverage the insight of Danihelka et al. (2022) that the objective of search at the root can be modeled as a *known-budget, simple-regret, best action identification* problem. This suggests using an algorithm such as SH which optimizes for this objective at the root directly and uses SMCTS as a subroutine to get samples of the action values at the root.

SH is a natural choice: in addition to being successful in MCTS it carries additional potential for SMC/TS: (i) it has additional variance-reducing effects compared to the SMC/TS baselines. (ii) It addresses the remaining effects of path degeneracy at the root. (iii) It is natural to use with particles, remaining fully parallelizable across particles

and maintaining the same runtime and space complexity as SMC/TS. (iv) In SMCTS the evaluation of actions at the root are stationary across iterations, unlike MCTS. This better fits the assumptions under which SH was designed and provides optimality guarantees (see Appendix A.5 and (Karnin et al., 2013)). We describe the novel Sequential-Halving Sequential-Monte-Carlo Tree Search algorithm, or *Twice Sequential Monte Carlo Tree Search* (**TSMCTS**) and motivate points (i-iii) in more detail, below.

Given a starting state $s_1 := s$ a number of particles $N$, depth budget $T$ and a number of starting actions to search at the root $m_1$ we have a total search budget (number of model expansions) $B = NT$, the particle budget multiplied by the depth budget. First, the algorithm computes the number of iterations: $\log_2 m_1$. The total compute budget is divided equally across iterations $B_i = NT/\log_2 m_1$. At each iteration, $i = 1, \ldots, \log_2 m_1$ each action $a_j \in A_i$ in the set of actions to search this iteration $A_i$ is searched independently in parallel from the root with an equal search budget $B_i^j = B_i/m_i$.

To preserve the parallelizability properties of SMC a constant $N$ particles are divided across the actions each iteration. To achieve this, we assign $N/m_i$ particles per-action per-iteration (we assume for simplicity that $m_i$ divides $N$ and otherwise round for a total particle budget of $N$ at each iteration). This results in the number of particles per-action per-iteration doubling every iteration $N_{i+1} = 2N_i$, as the number of actions halves $m_{i+1} = m_i/2$ maintaining the total number of particles per iteration constant. To maintain the total search budget of the algorithm $B$ the search horizon per iteration is reduced $T_{SH} \leq T$:

$$T_{SH} = \frac{B_i}{N/m_i} = \frac{NT}{m_i \log_2 m_1} \frac{m_i}{N} = \frac{T}{\log_2 m_1} \leq T. \quad (23)$$

This acts as an additional variance reducing mechanism, as SMC's variance increases with search depth and $T_{SH} \leq T$.

At the first iteration $i = 1$ the set of actions to search $|A_1| = m_1$ is sampled from $\pi_\theta(s_1)$ without replacement. In continuous action domains, actions are simply sampled from $\pi_\theta$. In discrete action spaces, to approximate sampling without replacement we follow the approach of Danihelka et al. (2022) which uses the Gumbel-top-k trick (Kool et al., 2019). Noise from the Gumbel distribution $(g \in \mathbb{R}^{|\mathcal{A}|}) \sim \text{Gumbel}(0)$ is added to the policy $\pi(s_1) \propto \exp(\log \pi_\theta(s_1) + g)$ and $A_1$ is taken according to:

$$i = 1 : A_1 = \underset{a \in A}{\arg \text{top}} \left( \pi(s_1), m_1 \right). \quad (24)$$

At each iteration $i \geq 1$ each action $a \in A_i$ is searched with SMCTS with $N_i$ particles, each action independently in parallel. This prevents the remaining effect of path degeneracy (see Appendix A.7 for more detail). The set $A_{i+1}$ of actions

to search in iteration $i + 1$ is chosen with:

$$i \geq 1 : A_{i+1} = \underset{a \in A_i}{\arg \text{top}} \left( \mathcal{I}(\pi, Q_i^{SH})(s_1), m_{i+1} \right). \quad (25)$$

Each iteration $i$, SMCTS is used to evaluate the next state $s_2 \sim P(s_1, a)$, $V_i^{SMCTS}(s_2) := V_{search}(s_2)$ for each action $a \in A_i$ in parallel. The value is used to compute the searched action's value: $Q_i(s_1, a) = r_i(s_1, a) + \gamma V_i^{SMCTS}(s_2)$. $Q_i$ is a lower-variance estimator than $Q_{i-1}$ for all actions visited this iteration, because the search budget per action doubles each iteration. As the variance of SMC scales with $1/N$ (Chopin & Papaspiliopoulos, 2020, Theorem 1) the variance-minimizing aggregation across iterations is the *inverse-variance weighting* (Hartung et al., 2011). That is, it weighs each $Q_j(s_1, a)$ by $N_j(a)$, the number of particles assigned to action $a \in A_i$ at previous iterations $j$:

$$Q_i^{SH}(s_1, a) = \frac{1}{\sum_{j=1}^{i} N_j(a)} \sum_{j=1}^{i} N_j(a) Q_j(s_1, a). \quad (26)$$

This results in an *additional* variance-reduction mechanism, which minimizes the variance of the value-maximizing actions: the actions that are the most important for action selection and policy improvement. For actions not searched at iteration $i$ the value estimate $Q_i(s_1, a)$ will be multiplied by $N_i(a) = 0$ and thus the actual value does not matter, so define $\forall a \notin A_i : Q_i(s_1, a) := 0$. This can be tracked with a running weighted average, maintaining the original space complexity. A detailed derivation of Equation 26 and discussion of the variance reduction mechanisms are provided in Appendices A.4 and A.6 respectively. TSMCTS returns:

$$\pi_{search}(s) = \mathcal{I}(\pi_\theta, Q_{\log_2 m_1}^{SH})(s), \quad (27)$$

$$V_{search}(s) = \sum_{a \in A_1} Q_{\log_2 m_1}^{SH}(s, a) \pi_{search}(a|s). \quad (28)$$

We summarize TSMCTS in Algorithm 3. TSMCTS maintains the space and runtime complexity of RL-SMC (see Appendix A.8). For implementation details see Appendix B.

Action selection is done by sampling from the improved policy $a \sim \pi_{search}(s)$ (deterministically during evaluation $a = \arg\max_{b \in A_1} \pi_{search}(b|s)$). The improved policy $\pi_{search}(s)$ is used to train the policy $\pi_\theta$ with cross entropy loss and the value estimate $V_{search}(s)$ to compute value targets to train the critic $v_\phi$ with MSE loss, as is standard for RL algorithms which use search (Algorithm 4).

One of the strengths of RL-SMC and T/SMCTS is that they allow for any greedification operator, enabling the methods to advance with novel, improved greedification operators. To keep comparison against baselines as direct as possible we've used the operator $\mathcal{I}_{GMZ}$ for both search and policy improvement at the root, which has become the popular choice (Danihelka et al., 2022; de Vries et al., 2025).

## 6. Related Work

SMC has been used in RL and more generally MDP solving for a variety of purposes (Lazaric et al., 2007; Hoffman et al., 2007; Le et al., 2018; Abdulsamad et al., 2025). Our focus in this section is on related work in the area of SMC for search in RL. Multiple works build upon Piché et al. (2019)'s derivation of SMC for search. Macfarlane et al. (2024) used SMC explicitly for policy improvement. Lioutas et al. (2023) extend the proposal distribution with a $q_\phi$ critic, to direct the mutation step towards more promising trajectories. de Vries et al. (2025) develop TRT SMC, which extends the SMC search further with trust-region optimization methods and additionally address terminal states with *revived resampling*. These contributions are orthogonal to ours and natural to incorporate into RL-SMC and TSMCTS (Figure 2). de Vries et al. (2025) also address path degeneracy by maintaining the *last* return observed for each root action preventing the collapse of the improved policy to a delta distribution. In contrast, SMCTS aggregates *all* returns observed during the search. This addresses path degeneracy in the same manner as well as reduces variance, as we demonstrate in the next section (Figure 1, right and center respectively). We include a brief summary of previous work on parallelizing MCTS and related challenges in Appendix A.8.

## 7. Experiments

The objective of this work is to improve SMC as a search algorithm for policy improvement in RL with our novel method TSMCTS. Specifically, we expect to see: (i) better sample efficiency through better policy targets. (ii) Improved capacity to *scale* with sequential compute, through (iii) significant variance reduction and (iv) mitigation of the effects of path degeneracy at the root. To demonstrate the advantages of TSMCTS over previous SMC variants we use the same experimental setup established by Macfarlane et al. (2024) and updated by de Vries et al. (2025) in the empirical evaluation. This setup contains a mix of discrete sparse-reward, single-goal and dense reward environments from Jumanji (Bonnet et al., 2024) and classic continuous control environments from Brax (Freeman et al., 2021).

We begin by evaluating *scaling with sequential compute*, *variance reduction* and *mitigation of the effects of path degeneracy at the root* in Figure 1. We compare the expected return of model based agents (Algorithm 4) which use SMC variants for policy improvement: (i) **TSMCTS** (ours), (ii) **SMCTS** (ours) and (iii) the **SMC baseline** which was used by SPO for policy improvement (CAI-SMC with $\mu = \pi_\theta$). Except for the novel contributions presented in this paper (T/SMCTS) our implementation of the agents is the same as that of de Vries et al. (2025), including hyperparameters. All agents search with the true dynamics model in the

AZ/SPO/TRT SMC manner. SMC and therefore T/SMCTS are agnostic to discrete / continuous actions, see Appendix B. All agents can be viewed as *SPO/TRT SMC variants* (using SMC for policy improvement through search with the true model) and more generally as *AZ variants* (policy improvement through search with the true model).

We plot: **(i)** scaling with sequential compute (increasing depth budget $T$, **left**). Performance is summarized as area-under-the-curve (AUC) for the evaluation returns during training normalized per environment and aggregated across environments. **(ii)** Variance of the root estimator (**center**). The variance is measured over the prediction of the root estimator for each planner $\mathbb{V}[V_{search}(s)] = \mathbb{V}[\sum_{a \in A} \pi_{search}(a|s)Q_{search}(s,a)]$ (where $A$ is the set of actions searched by the respective search algorithm). **(iii)** Policy collapse at the root (target degeneracy) as a measure for path degeneracy (**right**). Target degeneracy is measured as the number of actions active in the policy target. For additional details see Appendix D. In the variance and path degeneracy experiments we include an SMC variant (**TRT SMC**) which uses the path degeneracy mitigation mechanism proposed by de Vries et al. (2025). This, to demonstrate that while this mechanism mitigates path degeneracy in the same manner as SMCTS it addresses variance not as well. We demonstrate that the other contributions of TRT SMC are orthogonal to ours in Figure 2.

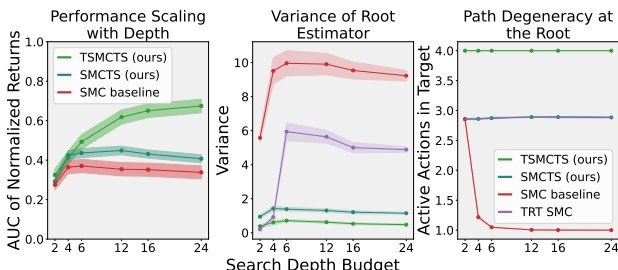

*Figure 1.* **Left:** Performance scaling with depth (*higher is better*), averaged across environments, particle budgets of $4, 8, 16$ and $10$ seeds. **Center:** Variance of the root estimator vs. depth (*lower is better*). **Right:** The number of actions active in the policy target (*higher is better, constant is better*). Center and right are averaged across states and particle budgets $4, 8, 16$ and $5$ seeds, in Snake. Mean and $95\%$ Gaussian CI for all curves.

The SMC baseline *degrades* with additional depth (left) due to high variance (center) and path degeneracy (right). SMCTS is able to reduce much of the variance (center) and performs better than the SMC baseline although it does not scale better with compute (right). On the other hand, **TSMCTS significantly outperforms the SMC baseline even at the depth which maximizes the SMC baseline's performance ($T = 6$) and is the only SMC variant that *scales* with increased compute (left). TSMCTS further reduces estimator variance significantly (center).** All variants other than baseline successfully prevent policy collapse at the root (right). **TSMCTS additionally provides more**

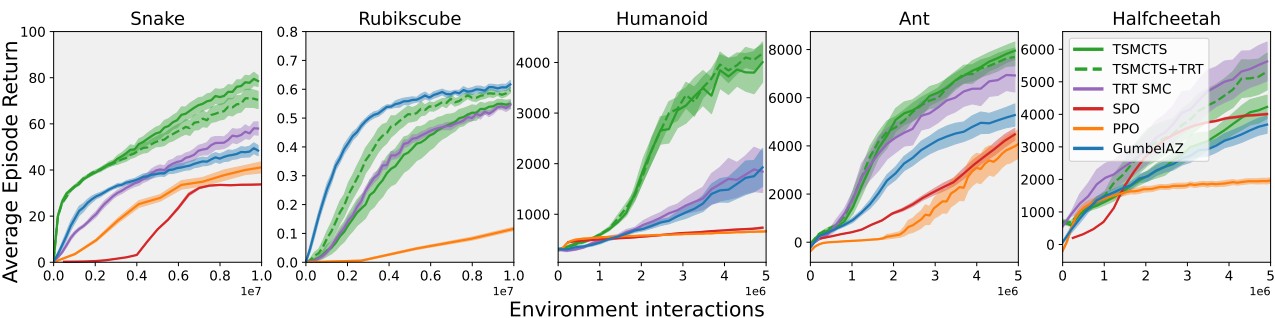

Figure 2. Averaged returns vs. environment interactions. 95% Gaussian CIs across 20 seeds.

**informed policy targets by searching a constant $m_1 = 4$ actions.** TRT SMC and SMCTS however are limited by the entropy of the prior policy: the policy has high probability for only two or three actions in most states despite the size of the action space being 4 in this environment and thus an average of only 2.8 actions are searched.

Next, we compare the TSMCTS based agent to popular baselines which use search for policy improvement: **SPO**, the full **TRT SMC** and **GumbelAZ**, an AZ agent using GumbelMCTS (Danihelka et al., 2022). GumbelAZ is extended to continuous environments in the manner of SampledMZ (Hubert et al., 2021) (see Appendix B for more detail). As discussed in Section 6, two of the three contributions of TRT SMC are orthogonal to ours. To demonstrate this we include a **TSMCTS + TRT** agent which incorporates these contributions to the backbone of TSMCTS. We include **PPO** (Schulman et al., 2017) for reference performance of a popular model-free baseline.

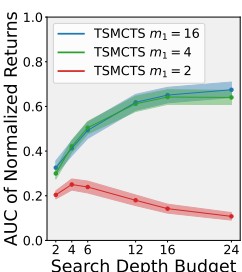

Figure 3. Performance scaling with depth normalized across environments (higher is better, increasing is better). Mean and 95% Gaussian CI across environments, particle budgets 4, 8, 16 and 10 seeds per combination.

The SMC-based variants (TSMCTS, TRT SMC, SPO) use $N = 4$ particles, the configuration used by TRT SMC and $T = 6$ which was the best performing depth for the SMC baseline in our experiments (Figure 1 left). For MCTS, $B = 24$ was used, to equate the compute used by MCTS and SMC variants across DNN forward passes, a popular choice in prior work (Macfarlane et al., 2024; de Vries et al., 2025) which does not take into account SMC's improved facility for parallelism and thus favors MCTS. We use the implementation of de Vries et al. (2025) for all baselines except for SPO which uses the original implementation (Toledo, 2024) in the environments in which it was implemented. We emphasize that TSMCTS, TRT SMC and GumbelAZ agents differ *only* in the search procedure used (TSMCTS, TRT SMC and GumbelMCTS,

respectively). The results are presented in Figure 2.

**In all environments TSMCTS-based agents significantly outperform all baselines or performs comparably, with the exception of RubiksCube where it matches the policy learned by MCTS at the end of training**. TSMCTS can benefit from the contributions of TRT in environments where they are beneficial (RubiksCube, HalfCheetah). In Figure 2 in the Appendix we ablate TSMCTS and SMCTS, by comparing AZ agents which use TSMCTS, SMCTS and baseline SMC for search, respectively. **TSMCTS significantly outperforms the SMC baseline in all environments in this on-equal-grounds comparison.** SMCTS does not outperform TSMCTS significantly in this domain with this low $T = 6$ depth budget, but in several environments (RubiksCube, Ant, HalfCheetah) it does not under-perform, as well.

In Figure 3 we investigate the effect of the hyperparameter $m_1$, the number of actions to search, on the performance of the agent. The effect appears negligible for sufficiently large $m_1 \geq 4$, with a possible exception of better performance for larger $m_1$ at high depth budgets. This would be expected, as increasing $m_1$ acts as a variance reduction mechanism, since the *effective search depth* $T_{SH} = \frac{T}{\log_2 m_1}$ reduces with the number of actions to search $m_1$ (Equation 23). In Figure 4 in Appendix C we include additional results evaluating runtime performance. TSMCTS demonstrates a small increase in compute overhead compared to the SMC baseline (both of which are implemented in JAX) and demonstrates significantly better runtime-efficiency than GumbelMCTS using its official implementation (DeepMind et al., 2020) (which is implemented in JAX as well).

## 8. Conclusions

We presented Twice Sequential Monte Carlo Tree Search (TSMCTS), a novel search algorithm based in Sequential Monte Carlo (SMC) for policy improvement in Reinforcement Learning (RL). TSMCTS builds upon our formulation of SMC for search in RL which extends prior work's (Piché et al., 2019; Macfarlane et al., 2024; de Vries et al., 2025) beyond the framework of Control As Inference (see (Levine,

2018)). TSMCTS incorporates mechanisms from Monte Carlo Tree Search (Świechowski et al., 2023) and Sequential Halving (Karnin et al., 2013) to mitigate the high estimator variance and path degeneracy problems of SMC, while maintaining SMC's beneficial parallelization, runtime and space complexity properties. In experiments across discrete and continuous environments TSMCTS outperforms the SMC baseline as well as a popular modern version of MCTS (GumbelMCTS, (Danihelka et al., 2022)) as a search-based policy improvement operator. In contrast to the SMC baseline, TSMCTS scales favorably with sequential compute, demonstrates lower estimator variance and mitigates the effects of path degeneracy at the root.

## Acknowledgments

Research reported in this work was partially or completely facilitated by computational resources and support of the Delft AI Cluster (DAIC) at TU Delft (RRID: SCR_025091), but remains the sole responsibility of the authors, not the DAIC team.

## Impact Statement

This paper presents work whose goal is to advance the field of Machine Learning. There are many potential societal consequences of our work, none which we feel must be specifically highlighted here.

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

## A. Theoretical Results

### A.1. RL-SMC is a policy improvement operator

*Proof.* Given exact evaluation $Q^\pi$, true environment model $P, R$, a starting state $s_1$ and infinitely many particles $N \to \infty$, the SMC target policy $\pi_{SMC}^T$ at final step $T$ produces the following distribution over trajectories:

$$p_t(\tau_T) = p_t(s_1, a_1, \ldots, s_T, a_T) = \pi'(a_1|s_1)\Pi_{i=1}^{T-1}P(s_{i+1}|s_i, a_i)\pi'(a_{i+1}|s_{i+1}) \tag{29}$$

The distribution $p(\tau_T)$ is the distribution induced by the policy $\pi'$ for all states $s_{1,\ldots,T}$ and by the policy $\pi$ for all other states, by definition. We have:

$$V^\pi(s_1) \leq \mathbb{E}_{\pi'}[Q^\pi(s_1, a_1)] \tag{30}$$
$$= \mathbb{E}_{\pi',P}[r_1 + \gamma V^\pi(s_2)] \tag{31}$$
$$\leq \mathbb{E}_{\pi',P}[r_1 + \gamma Q^\pi(s_2, a_2)] \tag{32}$$
$$\leq \mathbb{E}_{\pi',P}[r_1 + \gamma r_2 + \gamma^2 Q^\pi(s_3, a_3)] \tag{33}$$
$$\leq \ldots \tag{34}$$
$$\leq \mathbb{E}_{\pi',P}[r_1 + \cdots + \gamma^{T-2}r_{T-1} + \gamma^{T-1}Q^\pi(s_T, a_T)] \tag{35}$$
$$= V^{\pi_{SMC}^T}(s_1) \tag{36}$$

Equation 30 holds by definition of $\pi'$ produced from a greedification operator. Note that actions $a_1, a_2, \ldots$ are all sampled from $\pi'(s_1), \pi'(s_2), \ldots$ respectively, as the expectation is with respect to $\pi'$ at all steps. Equation 31 holds by definition of the value. Equation 32 holds because $\mathbb{E}_{\pi'}[Q^\pi(s_2, a_2)] \geq V^\pi(s_2)$ by definition of $\pi'$. Equation 33 is the two-step expansion following the same argumentation, and respectively Equation 35 is the multi-step expansion, which is the definition of the value of the policy $\pi_{SMC}^T$.

$\square$

### A.2. Deriving CAI-SMC in RL-SMC

The importance sampling weights of CAI-SMC derive as follows (see (Piché et al., 2019) for more detail):

$$u_t(\tau_t \mid \tau_{t-1}) = P(s_t \mid s_{t-1}, a_{t-1})\pi_\theta(a_t \mid s_t), \tag{37}$$
$$p_t(\tau_t \mid \tau_{t-1}) \propto P(s_t \mid s_{t-1}, a_{t-1})\mu(a_t \mid s_t)\mathbb{E}_{s_{t+1}|s_t,a_t}\big[\exp(A_{\text{soft}}(s_t, a_t))\big], \tag{38}$$
$$w_t^n = w_{t-1}^n \frac{p_t(\tau_t^n \mid \tau_{t-1}^n)}{u_t(\tau_t^n \mid \tau_{t-1}^n)} \propto w_{t-1}^n \frac{\mu(a_t^n \mid s_t^n)}{\pi_\theta(a_t^n \mid s_t^n)}\mathbb{E}_{s_{t+1}^n|s_t^n,a_t^n}\big[\exp(A_{\text{soft}}(s_t^n, a_t^n, s_{t+1}^n))\big], \tag{39}$$

Denote:

$$\pi'(a_t^n \mid s_t^n) = \mu(a_t^n \mid s_t^n)\mathbb{E}_{s_{t+1}^n|s_t^n,a_t^n}\big[\exp(A_{\text{soft}}(s_t^n, a_t^n, s_{t+1}^n))\big], \tag{40}$$

the soft-advantage based greedification with respect to the reference policy $\mu$. We have:

$$w_t^n \propto w_{t-1}^n \frac{\mu(a_t^n \mid s_t^n)}{\pi_\theta(a_t^n \mid s_t^n)}\mathbb{E}_{s_{t+1}^n|s_t^n,a_t^n}\big[\exp(A_{\text{soft}}(s_t^n, a_t^n, s_{t+1}^n))\big] = w_{t-1}^n \frac{\pi'(a_t^n \mid s_t^n)}{\pi_\theta(a_t^n \mid s_t^n)} \tag{41}$$

Which recovers RL-SMC.

### A.3. Policy Improvement with the value of an improved policy

Define the value $Q^{\pi'}$ of the improved policy $\pi'$ with respect to a reference policy $\pi$ such that $V^{\pi'}(s) \geq V^\pi(s), \forall s \in \mathcal{S}$ and there exists at least one state where the inequality is strict, unless $\pi$ is already the optimal policy (e.g. *policy improvement*, see Section 2). We will prove that Greedification $\pi'' = \mathcal{I}(\pi, Q^{\pi'})$ with respect to $\pi, Q^{\pi'}$ produces policy improvement. That is, that $V^{\pi''}(s) \geq V^\pi(s), \forall s \in \mathcal{S}$ and there exists at least one state where the inequality is strict, unless $\pi$ is already the optimal policy (Theorem 2).

*Proof.* Since $\pi'$ is an improved policy, we are given that $Q^{\pi'}(s,a) \geq Q^{\pi}(s,a), \forall (s,a) \in \mathcal{S} \times \mathcal{A}$. Therefore:

$$\forall s \in \mathcal{S}: \quad V^{\pi}(s) = \mathbb{E}_{\pi}[Q^{\pi}(s,a)] \tag{42}$$

$$\leq \mathbb{E}_{\pi}[Q^{\pi'}(s,a)] \tag{43}$$

$$\leq \mathbb{E}_{\pi''}[Q^{\pi'}(s,a)] \tag{44}$$

$$= V^{\pi''}(s) \tag{45}$$

The first equation holds by definition. Equation 43 holds because $Q^{\pi'}(s,a) \geq Q^{\pi}(s,a), \forall(s,a) \in \mathcal{S} \times \mathcal{A}$. Equation 44 holds by definition of greedification. That is, by definition $\pi''$ maximizes $Q^{\pi'}$ more than $\pi$. Finally, Equation 45 is the definition of the value.

By greedification, unless $\pi$ is already an optimal policy (a greedy policy with respect to the value of an optimal policy), there is at least one state where the inequality $V^{\pi}(s) < V^{\pi''}(s)$ is strict:

$$\exists s \in \mathcal{S}: \quad V^{\pi}(s) = \mathbb{E}_{\pi}[Q^{\pi}(s,a)] \tag{46}$$

$$\leq \mathbb{E}_{\pi}[Q^{\pi'}(s,a)] \tag{47}$$

$$< \mathbb{E}_{\pi''}[Q^{\pi'}(s,a)] \tag{48}$$

$$= V^{\pi''}(s) \tag{49}$$

$$\square$$

**Lemma 1.** *A mixture of improved policies is an improved policy.*

*Proof.* For any policy $\pi$ and improved policies $\pi_1', \ldots, \pi_n', n < \infty$, such that $\forall s \in S, i = 1, \ldots, n: V^{\pi_i'}(s) \geq V^{\pi}(s)$ and $\exists s \in S$ where the inequality is strict.

For any weights $\alpha_1, \ldots, \alpha_n$ such that $\forall i = 1, \ldots, n: \alpha_i \geq 0$ and $\sum_{i=1}^{n} \alpha_i = 1$ (any mixture), define the mixture policy:

$$\pi'(s) = \alpha_1 \pi_1'(s) + \alpha_2 \pi_2'(s) + \cdots + \alpha_n \pi_n'(s) \tag{50}$$

Due to the linearity of the expectation operator and the Bellman equation:

$$V^{\pi}(s) = \mathbb{E}_{\pi,P}\Big[ \sum_{t=1}^{H} \gamma^{t-1} R(s_t, a_t) \big| s_1 = s \Big], \tag{51}$$

we have:

$$V^{\pi'}(s) = \alpha_1 V^{\pi_1'}(s) + \alpha_2 V^{\pi_2'}(s) + \cdots + \alpha_n V^{\pi_n'}(s) \geq V^{\pi}(s) \tag{52}$$

$$\square$$

### A.4. Deriving the value update in TSMCTS

In MCTS, the value at each node $s$ equals the average of all $M$ returns observed through this node (Świechowski et al., 2023). This is because the variance of the estimator is expected to reduce with $1/M$, the number of visitations. This also holds in SMC, where for large number of particles $N$, the error behaves approximately Gaussian with variance proportional to $1/N$ (Chopin, 2004). For this reason, we rely on the same idea in TSMCTS.

At each iteration $i$ of TSMCTS the value estimate $Q_i(s_1, a)$ was computed using $N_i(a)$ particles per action. If we assume that each individual return has the same variance, the variance of this value estimate is proportional to $1/N_i(a)$. Therefore, according to inverse-variance weighting (Hartung et al., 2011), the contribution of this value estimate to the total average should be $N_i(a)$.

While it is unclear as to what extent this assumption holds in practice, MCTS was designed under the same assumptions (Kocsis & Szepesvári, 2006), and has been extremely successful even when the mechanism responsible for this assumption -

the evaluation of nodes using rollouts in the model - was replaced by a value DNN $v_\phi$ (Silver et al., 2016; 2018), for which it is unclear whether this assumption holds at all. Inspired by the success of MCTS, we design the value backpropagation of TSMCTS under the same assumption. A more general point of view would be that this update is expected to perform well in settings where this assumption is not far from the truth, and can be improved given access to a more accurate estimate of the variance. This is generally an active area of research (Oren et al., 2025a), and we leave a redesign of the backpropagation mechanism given access to more accurate variance predictions - applicable to MCTS as well - to future work.

Equation 26 (provided below again for readability) formulates this weighted average: it sums across the total number of iterations $\log_2 m_1$. For each iteration $i$, it multiplies $Q_i(s_1, a)$ by the weight $N_i(a)$. Finally, it normalizes the sum by $\sum_{i=j}^{i} N_j(a)$, the total number of particles "visiting" the action at the root up to iteration $i$:

$$\forall a \in M_1: \quad Q_i^{SH}(s_1, a) = \frac{1}{\sum_{j=1}^{i} N_j(a)} \sum_{j=1}^{i} N_j(a) Q_j(s_1, a)$$

Where $Q_j = r_j(s_1, a) + \gamma V_j^{SMCTS}(s_2^j)$ (when the model $R(s, a), P(s'|s, a)$ is stochastic, in general the conditioning need only be on $(s_1, a)$, and thus $V_j^{SMCTS}(s_2^j)$ and $r_j(s_1, a)$ may be unique to iteration $j$).

### A.5. Non/Stationary evaluations

MCTS is motivated from the perspective of *multi-armed bandits* (Kocsis & Szepesvári, 2006). From the RL perspective, this models the problem of action selection at each state $s_t$ during search as reasoning over evaluations $q_i(s_t, a), \forall a \in A$. MCTS assumes that the returns $q_i(s, a)$ estimated for the action chosen at step $i$ at state $s$ is sampled from some distribution with some mean $Q_i(s, a)$ and variance $\sigma^2$. This setup diverges from that of most bandit algorithms (such as SH) with the assumption that the distribution of evaluations has *non-stationary* means $Q_i(s, a)$ (e.g. the means depend on the step $i$). In contrast, in TSMCTS the means of the returns for each action $q_i(s, a) = r_i(s, a) + \gamma V_i^{SMCTS}(s_i')$, where $r_i(s, a) \sim R(s, a), s_i' \sim P(s'|s, a)$, *are* stationary: $q_i(s, a) \approx Q^{\pi_{SMCTS}^T}(s, a)$ ($\pi_{SMCTS}^T$ being the mixture policy described in Section 4, which is not dependent on $i$). This better aligns with the stationarity assumed by SH, under which the almost-optimality guarantees of the algorithm are established.

### A.6. Variance reduction

Throughout this work, we describe different mechanisms that reduce variance in TSMCTS compared to the SMC framework TSMCTS is built upon. In this section we will describe and motivate each mechanism in more detail. We begin with an overall motivation for variance minimization.

Variance minimization is a fundamental objective in statistical estimation, as the quality of an estimator is typically assessed through its mean squared error (MSE) (Geman et al., 1992). The MSE admits a standard decomposition into the squared bias and the variance,
$$\text{MSE} = \text{Bias}^2 + \text{Var}.$$

While bias captures systematic deviation from the true quantity, variance reflects the sensitivity of the estimator to fluctuations in the data. Minimizing variance - without changing the bias - therefor reduces to minimizing estimation error. We proceed to describe each variance-reducing mechanism in chronological order.

**Backpropagation in SMCTS** The running means $\bar{Q}_t(s_1, a_1)$ maintained through backpropagation in SMCTS decompose into:

$$\bar{Q}_t(s_1, a_1) = \frac{1}{t} \sum_{i=1}^{t} Q_i(s_1, a_1) = \frac{1}{t} \sum_{i=1}^{t} \sum_{j=1}^{N} \bar{w}_i^j \mathbb{1}_{a_1^{(i,j)} = a_1} \sum_{k=1}^{i} \gamma^{k-1} r_k^j + \gamma^t V^{\pi_\theta}(s_{i+1}^j). \tag{53}$$

$\bar{Q}_t(s_1, a_1)$ is a reduced variance estimator compared to $Q_t$ for two reasons.

(i) Consider the bootstrapped return:

$$Q_t(s_1, a_1) = \sum_{i=1}^{t} \gamma^{i-1} r_i + \gamma^t V^{\pi_\theta}(s_{t+1}). \tag{54}$$

For any $t < h$, the estimator $Q_t(s_1, a_1)$ terminates earlier and bootstraps from $V^{\pi_\theta}$ sooner. Extending the horizon from $t$ to $h$ replaces a single (deterministic) bootstrap term with additional possibly-random rewards and transitions. This introduces additional stochasticity (unless the policy, reward and transition dynamics are all deterministic). Consequently, $\mathrm{Var}(Q_t(s_1, a_1)) \leq \mathrm{Var}(Q_h(s_1, a_1))$, reflecting the classical result that Monte Carlo returns (larger $h$) have higher variance than temporally shorter, bootstrapped estimates (smaller $t$) (Sutton & Barto, 2018).

(ii) Even when the policy, reward and transition dynamics *are* all deterministic, mixing returns of different horizons can result in variance reduction. Any errors in the value prediction $v_\phi$ that are I.I.D. will average out in the empirical average (weighted or otherwise) $\frac{1}{N} \sum_{t=1}^{N} \sum_{i=1}^{t} \gamma^{i-1} r_i + \gamma^t v_\phi(s_{t+1})$, resulting in possible variance reduction even in the fully deterministic policy and MDP case.

We note that the value $V^{\pi_\theta}(s_{t+1})$ evaluated at any $s_{t+1}$ approximates the value of the policy $\pi_\theta$, not the policy $\pi_{SMC}^T$ which induces the trajectory $\tau(s_t)$. This analysis holds under the assumption that $\pi'$ does not increase variance compared to $\pi_\theta$.

**Repeatedly searching the same actions from the root in TSMCTS**   At each iteration $i$, TSMCTS searches a set of actions $A_{i+1} \subset A_i$. Since the actions are searched independently again from the root, the variances satisfy $\mathbb{V}[Q_i^{SH}(s_1, a)] < \mathbb{V}[Q_i(s_1, a)], \forall a \in A_i$. That is, the average *across* the value estimates of independent iterations is a lower variance estimate of the true value compared to each individual estimate, for the same reasoning as above.

**Increasing particle budget per searched action at the root in TSMCTS**   Under standard assumptions, increasing the number of particles in SMC algorithms reduces variance because the particle system provides an empirical average, and the variance of such Monte Carlo estimates decreases proportionally to the number of particles $1/N$, where $N$ is the number of particles (Chopin & Papaspiliopoulos, 2020).

**Searching for a shorter horizon**   TSMCTS trades off the depth of the search $T_{SH} < T$ for repeated search from the root. Reducing the depth of the search has two main effects: (i) It reduces the number of consecutive improvement (or search) steps. In a manner of speaking, the resulting policy is "less improved". (ii) It results in a lower variance estimator, as the variance grows in $t$ and $T_{SH} < T$ for all $m_1 > 2$.

### A.7. Addressing the Effects of Path Degeneracy

As discussed in Section 3, path degeneracy has the following detrimental effects on search for policy improvement: (i) Once paths have degenerated to one root action, the algorithm cannot benefit from additional search depth $T > h$. This is itself for two reasons: first, the policy at the root will not change with any additional search. This is addressed by reasoning over values rather than trajectories, resulting in the algorithm being able to benefit from search even just for one root action. Second, no other action will ever be searched, resulting in possibly to early a focus on one action, and preventing effective identification of the best action. This is addressed by SH, which at each iteration searches all actions $a \in A_i$ using a budget with optimality guarantees for best action identification. (ii) It results in a delta distribution policy target that is a crude approximation for any underlying improved policy $\pi_{search}(s_1)$ but an $\arg\max$. This is addressed by maintaining an aggregate across values and extracting policy improvement with respect to these values.

### A.8. Complexity Analysis

We include a brief runtime and space complexity analysis for MCTS and RL-SMC.

**MCTS complexity**   For a search budget $B$, MCTS conducts $B$ iterations. At each iteration $i$, MCTS conducts $d_i \leq B$ search steps, one expansion step, and then $d_i \leq B$ backpropagation steps along the nodes in the trajectory. $d_i$ denotes the depth of the leaf at step $i$. We can therefor bound the runtime complexity by $\mathcal{O}(B(B + B + 1)) = \mathcal{O}(B^2)$ operations. In regards to space complexity, MCTS construct a tree of size $B$, so the space required is of complexity $\mathcal{O}(B)$.

**RL-SMC complexity**   For $N$ particles and a depth $T$, the search budget of RL-SMC totals $NT = B$. RL-SMC executes a constant number of operations for $T$ steps for $N$ particles in parallel as well as *selection* (e.g. resampling). The cost of selection is $\mathcal{O}(N)$. With selection at every $t$, or in periods of $t$, the cost is thus $\mathcal{O}(NT) = \mathcal{O}(B) < \mathcal{O}(B^2)$. In terms of space, RL-SMC maintains only statistics about each particle, resulting in space complexity of $\mathcal{O}(N) \leq \mathcal{O}(B)$. Since RL-SMC is a generalization of (Piché et al., 2019)'s CAI-SMC, we conclude that CAI-SMC has the same space and runtime

complexity.

**SMCTS complexity**   For $N$ particles and a depth $T$, the search budget of SMCTS totals $NT = B$. At each step $i$, SMCTS conducts a constant number of additional operation: one running sum is maintained for each particle, and one running sum is maintained for each searched action at the root. In addition, SMCTS normalizes the weights at each $t$, resulting in a similar sequential runtime complexity of $\mathcal{O}(NT) < \mathcal{O}(B^2)$. SMCTS maintains statistics about $N$ particles, and also statistics about $M \leq N$ searched actions at the root. This results in space complexity of $\mathcal{O}(2N) = \mathcal{O}(N) \leq \mathcal{O}(B)$, the same space complexity as RL-SMC.

**TSMCTS complexity**   For $N$ particles and a depth $T$, the search budget of SMCTS totals $NT = B$. TSMCTS divides this budget across $\log_2 m_1$ iterations. At each iteration, TSMCTS executes SMCTS with depth $T/\log_2 m_1$, resulting in runtime complexity of $\mathcal{O}(N \log_2 m_1 \frac{T}{\log_2 m_1}) = \mathcal{O}(NT) < \mathcal{O}(B^2)$, the same as SMCTS and RL-SMC. In terms of space complexity, TSMCTS maintains statistics over $N$ particles, and $m_1 \leq N$ searched actions at the root, resulting in the same space complexity as RL-SMC and SMCTS, $\mathcal{O}(2N) = \mathcal{O}(N) \leq \mathcal{O}(B)$.

**Parallelizing MCTS**   Approaches to parallelize MCTS exist (Chaslot et al., 2008). These range from running *leaf parallelization* which performs multiple independent rollouts from the same newly expanded leaf node, improving evaluation accuracy but not accelerating tree growth. This of course is not applicable with modern MCTS methods which use a value DNN to expand leaves. *Search parallelization* runs MCTS in parallel across multiple states in multiple environments in parallel. This is the current norm for JAX based implementations, such as (DeepMind et al., 2020). *Root parallelization* launches multiple independent MCTS instances — each constructing its own search tree — and aggregates root-level statistics. This is in in direct competition over resources with *search parallelization*. Since it runs multiple trees for the same state, it reduces the number of independent states that can be searched in parallel, and thus slows down data gathering (number of environment interactions per search steps). *Tree parallelization* is the most akin to the parallelization of SMC: it shares a single MCTS tree among multiple workers, requiring synchronization mechanisms, such as local mutexes and virtual losses (a heuristic to prevent all workers from searching the same trajectory each iteration) to maintain consistency and avoid redundant exploration. This contrasts with the ease at which SMC parallelizes, as it is fundamentally a particle based method.

**A note on complexity in practice**   It is unlikely that all operations will have the same compute cost in practice. In search algorithms that use DNNs, it is often useful to think of two separate operation costs: model interactions, and DNN forward passes. Either of the two can often be the compute bottleneck, depending on the choice of model, DNN architecture, hardware etc. This motivates an equating for compute estimated in number of model expansions / DNN forward passes which is the same in both MCTS and the SMC based variants: $B$ for MCTS and $NT$ for the SMC variants, which is why previous work (Macfarlane et al., 2024; de Vries et al., 2025) as well as us, opted to compare MCTS and SMC variants with budgets $B = NT$.

## B. Implementation Details

**Targets and losses**   Our implementation for all search-based agents uses a $v_\phi$ critic and a prior policy $\pi_\theta$, targets $\pi_{search}(s_t)$ and $v_t$ which is computed using TD-$\lambda$ with bootstraps $V_{search}$ and a circular replay buffer $\mathcal{D}_{(n)}$. The value and policy are trained with the following losses:

$$\mathcal{L}(\theta) = \mathbb{E}_{(s_t, \pi_{search}(s_t)) \sim \mathcal{D}_{(n)}} \left[ -\mathbb{E}_{a \sim \pi_{search}(s_t)} \ln \pi_\theta(a|s_t) - c_{ent}\mathcal{H}[\pi_\theta(a|s_t)] \right], \tag{55}$$

$$\mathcal{L}(\phi) = \mathbb{E}_{(s_t, V_{search}(s_t)) \sim \mathcal{D}_{(n)}} \left[ (V_{search}(s_t) - v_\phi(s_t))^2 \right]. \tag{56}$$

**The training loop**   The RL training setup follows the popular approach in JAX: gather *batch size B* interaction trajectories of length *unroll length L* in parallel. The agent is then trained for $K$ *SGD update steps* with *SGD minibatch size* (see hyperparameters in Table 1) and the above losses. Following that, the agent proceeds to gather a additional data of size $LB$. The AdamW optimizer (Loshchilov & Hutter, 2019) was used with an $l_2$ penalty of $10^{-6}$ and a learning rate of $3 \cdot 10^{-3}$. Gradients were clipped using a max absolute value of 10 and a global norm limit of 10.

**Discrete vs. continuous action spaces**   The same losses are used to train the value and policy networks, irrespective of the type of action space. In continuous environments, the policy is a Gaussian policy, predicting mean and variance. In discrete

environments, the policy is trained to predict the log-probabilities for each action in the action space. To extend MCTS to continuous actions we follow the popular approach of SampledMZ (Hubert et al., 2021), which samples $C$ actions from the prior policy at each node in the search tree and treats $\{a_1, \ldots, a_C\}$ as a discrete action space.

**On / off policy considerations** In Algorithm 4 we formulate a simple *on-policy* search-based model-based RL algorithm. By on policy, we refer to algorithms that cannot (/should not) learn from data generated by any other policy but their current $\pi_{\theta_n}$. Such algorithms require, in principle, small replay buffers, otherwise they suffer from training targets becoming stale. This is a popular framework to use with JAX, which benefits from all data structures (the DNNs, environment implementation, replay buffer etc.) being saved in the GPU's RAM. Since the amount of RAM is often a bottleneck, it suggests using small replay buffers to start with, making on-policy training with small replay buffers natural. This is also the framework the agents were implemented in by de Vries et al. (2025) which we build on.

However, these algorithms are *not* generally limited to being on-policy. Schrittwieser et al. (2021) propose a simple and elegant approach to solve the apparent on-policy-ness of this family of algorithms: simply re-run the search process $\mathcal{P}$ on any state $s$ from the replay buffer, from scratch, to generate fresh targets / bootstraps $\pi_{search}(s), V_{search}(s)$. This allows the algorithms to learn well in principle from data generated from any policy, at the cost of additional compute spent on fresh calls to the search operator $\mathcal{P}$.

Pseudocode for the different algorithms is provided below.

---

**Algorithm 1** RL-SMC

---

**Require:** Number of particles $N$, depth $T$, model $P$, prior-policy $\pi_\theta$, policy improvement operator $I$, value function $Q^{\pi_\theta}$ and current state in the environment $s_{root}$.

1: Initialize particles $n \in N$, with $w_0^n = 1, s_1^n = s_{root}$,, ancestor identifier $\{j_0^n = n\}_{n=1}^N$ which identifies per particle which action $a_1 \in \mathcal{A}$ at the root it is associated with, selection period $M$ (As $M$ *decreases* variance *decreases* but path degeneracy *increases*, see (Chopin & Papaspiliopoulos, 2020)).

2: **for** $t = 1$ to T **do**

3:    *Mutation*: $\{a_t^n \sim \pi_\theta(a_t|s_t^n)\}_{n=1}^N, \quad \{s_{t+1}^n \sim P(\cdot|s_t^n, a_t^n)\}_{n=1}^N$.

4:    *Correction*: $\{w_t^n = w_{t-1}^n \frac{\pi'(a_t^n|s_t^n)}{\pi_\theta(a_t^n|s_t^n)}, \quad \pi'(s_t^n) = \mathcal{I}(Q^{\pi_\theta}, \pi_\theta)(s_t^n) \quad j_t^n = j_{t-1}^n\}_{n=1}^N$.

5:    **if** $t \mod M = 0$ **then**

6:      *Selection*: $\{(j_t^n, a_t^n, s_{t+1}^n)\}_{n=1}^N \sim \text{Multinomial}(N, \text{normalized } w_t), \quad \{w_t^n = 1\}_{n=1}^N$.

7:    **end if**

8: **end for**

9: Return $\{j_T^n, w_T^n\}_{n=1}^N$

---

---

**Algorithm 2** SMCTS

---

**Require:** Number of particles $N$, depth $T$, starting state for planning $s_{root}$, model $\mathcal{M} = (P, R)$, prior-policy $\pi_\theta$, value network $v_\phi$, improvement operators for search $\mathcal{I}_{search}$ and root $\mathcal{I}_{root}$.

1: Initialize particles $n \in N$, with $w_0^n = 1, s_1^n = s_{root}$, non-bootstrapped returns $R_0^n = 0$, ancestor identifiers $\{j_1^n = n\}_{n=1}^N$.

2: **for** $t = 1$ to $T$ across $n$ particles in parallel **do**

3:     *Mutation*: $a_t^n \sim \pi_\theta(s_t^n)$, $r_t^n \sim R(s_t^n, a_t^n)$, $s_{t+1}^n \sim P(s_{t+1}^n | s_t^n, a_t^n)$.

4:     If $t = 1$, maintain the set of $N$ root actions: $A_1 \leftarrow \{a_1^n\}_{n=1}^N$.

5:     Approximate state-action value: $Q(s_t^n, a_t^n) \leftarrow r_t^n + \gamma v_\phi(s_{t+1}^n)$.

6:     Compute the search policy: $\pi'(s_t^n) \leftarrow \mathcal{I}_{search}(\pi_\theta, Q)(s_t^n)$.

7:     *Correction:* Compute importance sampling weights (Equation 13): $w_t^n = w_{t-1}^n \frac{\pi'(a_t^n, s_t^n)}{\pi_\theta(a_t^n, s_t^n)}$.

8:     Update the non-bootstrapped returns: $R_t^n = R_{t-1}^n + \gamma^{t-1} r_t^n$.

9:     Normalize importance sampling weights *per action at the root* using their identifiers $j_t^n$:

$$\bar{w}_t^n = \frac{w_t^n}{\sum_{k=1}^N w_t^k \mathbb{1}_{j_t^n = j_t^k}}.$$

10:     Estimate $Q_t(s_{root}, \cdot)$ for initial actions $a_1^n \in A_1$ using $R_t^n$ and $v_\phi(s_{t+1})$:

$$Q_t(s_{root}, a_1^n) = \sum_{i=1}^N \bar{w}_t^i (R_t^i + \gamma^t v_\phi(s_{t+1}^i)) \mathbb{1}_{j_1^n = j_t^i}$$

11:     Update the running average $\bar{Q}_t(s_{root}, \cdot)$ where $Q_t(s_{root}, \cdot)$ is defined:

$$\bar{Q}_t(s_{root}, a_1^n) = \frac{(t-1)\bar{Q}_{t-1}(s_{root}, a_1^n) + Q_t(s_{root}, a_1^n)}{t}$$

12:     **if** $t \mod M = 0$ **then**

13:         *Selection*: $\{(j_t^n, a_t^n, s_{t+1}^n)\}_{n=1}^N \sim \text{Multinomial}(N, \text{normalized } w_t), \quad \{w_t^n = 1\}_{n=1}^N$.

14:     **end if**

15: **end for**

16: Compute improved policy $\pi_{search}(s_{root}) \leftarrow \mathcal{I}_{root}(\pi_\theta, \bar{Q}_T)(s_{root})$.

17: Compute the value of the improved policy across the set of root actions $A_1$:

$$V_{search}(s_{root}) \leftarrow \sum_{a \in A_1} Q_T(s_{root}, a) \pi_{search}(a | s_{root}).$$

18: Return $\pi_{search}(s_{root}), V_{search}(s_{root})$.

---

---

**Algorithm 3** TSMCTS

---

**Require:** Number of particles $N$, planning depth $T$, number of actions to search at the root $m_1$, model $\mathcal{M} = (P, R)$, policy network $\pi_\theta$, value network $v_\phi$, state in the environment $s_{root}$, Gumbel noise vector $g$ and greedification operators $\mathcal{I}_{search}, \mathcal{I}_{root}$.

1: Compute the per-iteration depth (Equation 23): $T_{SH} \leftarrow T/\log_2 m_1$.

2: Get $m_1$ starting actions (Equation 24): $A_1 = \{a_1, \ldots, a_{m_1}\} \leftarrow \arg\mathrm{top}(\pi_\theta(s_{root}) + g, m_1)$ or $A_1 \sim \pi_\theta(s_{root})$ in a continuous state space $\mathcal{S}$.

3: Initialize running sum of particles per action $N_0^{sum}(a) \leftarrow 0$ and running value sum per action $Q_0^{sum}(s_{root}, a) \leftarrow 0$, for all actions at the root $a \in A_1$.

4: Compute starting number of particles per action: $N_1 \leftarrow floor(N/m_1)$.

5: **for** $i = 1$ to $\log_2 m_1$ **do**

6:     **for** each action $a \in A_i$ in parallel **do**

7:         Sample $s_2 \sim P(\cdot|s_{root}, a)$, $r_i(s_{root}, a) \sim R(s_{root}, a)$.

8:         Search using SMCTS:

$$\_, V_i^{SMCTS}(s_2) \leftarrow \mathrm{SMCTS}(N_i, T_{sh}, s_2, \mathcal{M}, \pi_\theta, v_\phi, \mathcal{I}_{search}, \mathcal{I}_{root}).$$

9:         Approximate the value of action $a$: $Q_i(s_{root}, a) = r_i(s_{root}, a) + \gamma V_i^{SMCTS}(s_2)$.

10:        Update the running sums (or in place weighted averages) of particles and values of action $a$:

$$N_i^{sum}(a) \leftarrow N_{i-1}^{sum}(a) + N_i, \quad Q_i^{sum}(s_{root}, a) \leftarrow Q_{i-1}^{sum}(s_{root}, a) + N_i Q_i(s_{root}, a).$$

11:     **end for**

12:     Compute the current iteration's value estimate at the root (Equation 26):

$$\forall a \in A_i: \quad Q_i^{SH}(s_{root}, a) \leftarrow \frac{Q_i^{sum}(s_{root}, a)}{N_i^{sum}(a)} = \frac{1}{\sum_{j=1}^i N_j(a)} \sum_{j=1}^i N_j(a) Q_j(s_{root}, a).$$

13:     Update the number of actions to search: $m_{i+1} = m_i/2$.

14:     Update the actions to search (Equation 25): $A_{i+1} = \arg\mathrm{top}_{a \in A_i}(\mathcal{I}_{root}(\pi, Q_i^{SH})(s_1), m_{i+1})$

15:     Update the running number of particles per action: $N_{i+1} \leftarrow 2N_i$.

16: **end for**

17: Compute the final Q-estimate (Equation 26): $\forall a \in A_1: \quad Q_{\log_2 m_1}^{SH}(s_{root}, a) \leftarrow \frac{Q_{\log_2 m_1}^{sum}(s_{root}, a)}{N_{\log_2 m_1}^{sum}(a)}.$

18: Compute the improved policy $\pi_{search}$ using $\mathcal{I}_{root}$:

$$\forall a \in A_1: \; \pi_{search}(a|s_{root}) \leftarrow \mathcal{I}_{root}(\pi_\theta, Q_{\log_2 m_1}^{SH}), \quad \forall a \notin A_1: \; \pi_{search}(a|s_{root}) \leftarrow 0$$

19: And its value: $V_{search}(s_{root}) = \sum_{a \in A_1} \pi_{search}(a|s_{root}) Q_{\log_2 m_1}^{SH}(s_{root}, a)$.

20: Return $\pi_{search}(s_{root}), V_{search}(s_{root})$.

---

**Algorithm 4** Simplified Model Based RL Training Loop with Modular Search

---

**Require:** Search algorithm (planner) $\mathcal{P}$, neural networks $\pi_{\theta_1}, v_{\phi_1}$, replay buffer $\mathcal{D}_{(1)} = \emptyset$, environment's dynamics model
$\quad \mathcal{M} = (P, R)$ and budget parameters $B$.

1: **for** episode $n = 1$ to $N$ **do**
2: $\quad$ Sample starting state $s_1 \sim \rho$.
3: $\quad$ **for** step $t = 1$ in the environment to termination or timeout **do**
4: $\quad\quad \pi_{search}(s_t), V_{search}(s_t) \leftarrow \mathcal{P}(\pi_{\theta_n}, v_{\phi_n}, \mathcal{M}, B)(s_t)$.
5: $\quad\quad a_t \sim \pi_{search}(s_t)$.
6: $\quad\quad s_{t+1} \sim P(\cdot|s_t, a_t), \quad r_t \sim R(s_t, a_t)$.
7: $\quad\quad$ Append $(s_t, a_t, r_t, s_{t+1}, \pi_{search}(s_t), V_{search}(s_t))$ to buffer $\mathcal{D}_{(n)}$.
8: $\quad$ **end for**
9: $\quad$ Update policy params $\theta_{n+1}$ with SGD and CE loss on targets $\pi_{search}$ from $\mathcal{D}_{(n)}$.
10: $\quad$ Update value params $\phi_{n+1}$ with SGD and MSE loss on TD-$\lambda$ targets using $V_{search}$ from $\mathcal{D}_{(n)}$.
11: $\quad$ Set $\mathcal{D}_{n+1} = \mathcal{D}_n$.
12: **end for**

---

## C. Additional Experiments

**Runtime efficiency**  In Figure 4 we include a reference runtime comparison between the three search algorithms: baseline SMC, TSMCTS and GumbelMCTS. Runtime was estimated by multiplying training step with average runtime-per-step. TSMCTS induces a modest runtime increase over SMC for the same compute resources and compares very well to MCTS which has roughly twice the runtime cost as the SMC-based variants.

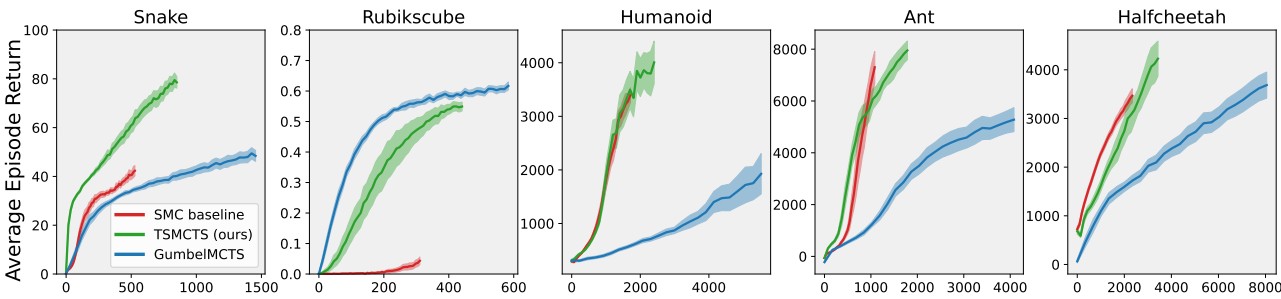

*Figure 4.* Averaged returns vs. runtime (seconds). Mean and 95% Gaussian CI across 20 seeds.

**Learning curves**  Below in Figure 5 we include the learning curves for the SMC baseline, SMCTS and TSMCTS for the configuration of Figures 2 and 4 (i.e. $T = 6$ and $N = 4$). Only TSMCTS significantly outperforms the SMC baseline in all domains.

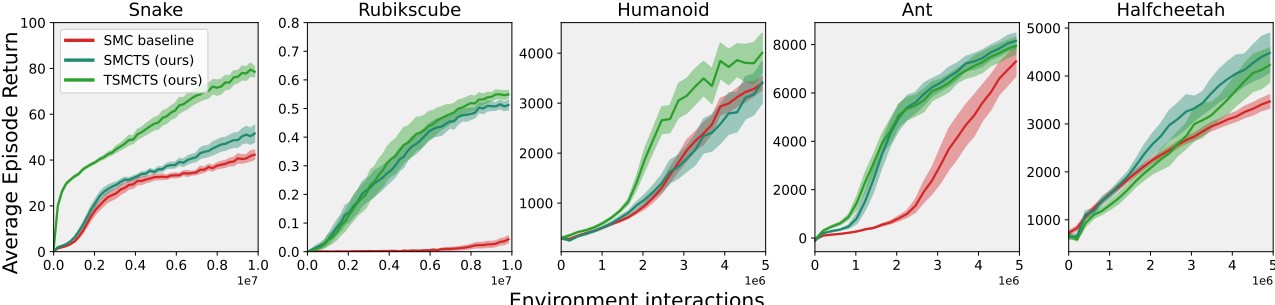

*Figure 5.* Averaged returns vs. environment interaction. Mean and 95% Gaussian CI across 20 seeds.

# D. Experiments Details

For the experiments, we build on the setup proposed by (de Vries et al., 2025), which we describe in more detail below.

**Environments**   We have used Jumanji's (Bonnet et al., 2024) Snake-v1 and Rubikscube-partly-scrambled-v0, as well as Brax's (Freeman et al., 2021) Ant, Halfcheetah and Humanoid.

**Compute**   All experiments were run on the (Delft AI Cluster (DAIC), 2024) cluster with a mix of Tesla V100-SXM2 32GB, NVIDIA A40 48GB, and A100 80GB GPU cards. Each individual run (seed) used 2 CPU cores and $\leq 6$ GB of VRAM.

**Wall-clock Training Time Estimation**   To estimate the training runtime in seconds (Figure 4), we used an estimator of the the runtime-per-step (total runtime divided by steps) and multiplied this by the current training step to obtain a cumulative estimate. This estimator should more robustly deal with the variations in hardware, the compute clusters' background load and XLA dependent compilation. Of course, estimating runtime is strongly limited to hardware and implementation and the results presented in Figure 4 should only be taken with that in mind.

**Normalization across environments**   The normalization across environments in Figure 1 was done as follows: per environment, the AUC were normalized with respect to minimum and maximum AUCs observed over all agents and seeds. The AUCs were then aggregated per agent across environments. Finally, mean-AUC with 95% Gaussian CI were computed on the aggregated AUCs.

**Variance and Path Degeneracy Estimation**   In Figure 1 center we plot the variance of the root estimator $V(s_0) = \sum_{a \in M} \pi_{improved}(a|s) Q_{search}(s, a)$ at the end of training as a function of depth for each method. $M$ is the number of actions over which the estimator maintains information (susceptible to path degeneracy). The variance is computed across $L = 128$ independent calls to each planner per seed at every state in an evaluation episode after training has completed in the Snake environment and then averaged across states and seeds. All planners used the same DNNs. The 95% Gaussian CI is then computed across seeds. Following (de Vries et al., 2025), for TRT-SMC and the SMC baseline we compute $Q_{search}$ as the TD-$\lambda$ estimator for each particle at the root for the *last* depth $t = T$. If multiple particles are associated with the same action at the root, the particle estimates are averaged. To address path degeneracy when all particles for a root action are dropped TRT-SMC saves the last TD-$\lambda$ estimate for each root action. For T/SMCTS we use $V_{T/SMCTS}$ respectively. In Figure 1 right we plot the number of actions at the root with which information is associated at the end of training, $M$, vs. depth. The number of active actions at the root is averaged across the $L$ calls to the search algorithm.

**Neural Network Architectures**   As specified by (de Vries et al., 2025), which are themselves adapted from (Bonnet et al., 2024) and (Macfarlane et al., 2024) (e.g. MLPs in all environments except Snake where a CNN followed by an MLP is used).

**Hyperparameters**   We've used the hyperparameters used by (Macfarlane et al., 2024) and (de Vries et al., 2025) for these tasks (when conflicting, we've used the parameters used by the more recent work (de Vries et al., 2025)). Except for the two new hyperparameters introduced by T/SMCTS no hyperparameter optimization took place. These new hyperparameters are (i) $m_1$, for which results are presented in Figure 3. (ii) The $\beta_{root}$ inverse-temperature hyperparameter of $\mathcal{I}_{GMZ}$ used by T/SMCTS to compute the improved policy at the root ($\mathcal{I}_{root}$). For $\beta_{root}$ we conducted a grid search with a small number of seeds across environments and values of $0.1^{-1}, 0.05^{-1}, 0.01^{-1}, 0.005^{-1}$. $\beta = 0.01^{-1}$ was overall the best performer. The $\beta_{search}$ hyperparameter is actually the same parameter as the *target temperature* used by the SMC baseline (see de Vries et al., 2025). We have not observed differences in performance across a range of parameters $\beta_{search}$ for TSMCTS and opted to use the same value as SMC.

Hyperparameters are summarized in Tables 1, 2, 3, 4, 5 and 6.

| Name | Value Jumanji | Value Brax |
|------|---------------|------------|
| SGD Minibatch size | 256 | 256 |
| SGD update steps | 100 | 64 |
| Unroll length (nr. steps in environment) | 64 | 64 |
| Batch-Size (nr. parallel environments) | 128 | 64 |
| (outer-loop) Discount | 0.997 | 0.99 |
| Entropy Loss Scale ($c_{ent}$) | 0.1 | 0.0003 |

*Table 1.* Shared experiment hyperparameters.

| Name | Value Jumanji | Value Brax |
|------|---------------|------------|
| Policy-Ratio clipping | 0.3 | 0.3 |
| Value Loss Scale | 1.0 | 0.5 |
| Policy Loss Scale | 1.0 | 1.0 |
| Entropy Loss Scale | 0.1 | 0.0003 |

*Table 2.* PPO hyperparameters.

| Name | Value Jumanji | Value Brax |
|------|---------------|------------|
| Replay Buffer max-age | 64 | 64 |
| Nr. bootstrap atoms (actions sampled) | 30 | 30 |
| Max depth | 16 | 16 |
| Max breadth | 16 | 16 |

*Table 3.* GumbelMCTS hyperparameters.

| Name | Value Jumanji | Value Brax |
|------|---------------|------------|
| Replay Buffer max-age | 64 | 64 |
| Selection (Resampling) period | 4 | 4 |
| Target temperature | 0.1 | 0.1 |
| Nr. bootstrap atoms | 30 | 30 |

*Table 4.* Shared SMC hyperparameters.

| Name | Value Jumanji | Value Brax |
|------|---------------|------------|
| (inner-loop) Retrace $\lambda$ | 0.95 | 0.9 |
| (inner-loop) Discount | 0.997 | 0.99 |
| (outer-loop) Value mixing | 0.5 | 0.5 |
| Estimation $\pi_{improved}$ | Message-Passing | Message-Passing |

*Table 5.* TRT-SMC variance ablation hyperparameters.

| Name | Value |
|------|-------|
| Root policy improvement operator ($\mathcal{I}_{root}$) | $\mathcal{I}_{GMZ}$ |
| Search policy improvement ($\mathcal{I}_{search}$) | $\mathcal{I}_{GMZ}$ |
| Root inverse temperature $\beta_{root}$ | $0.01^{-1}$ |
| Search inverse temperature $\beta_{search}$ | $0.1^{-1}$ |
| Number of actions to search at the root $m_1$ | 4 (Figures 2, 1 center and right subplots, 4, and 5), 16 (Figure 1, left) |

*Table 6.* SMCTS and TSMCTS hyperparameters.

