# OpenReview forum: "Twice Sequential Monte Carlo for Tree Search"
_ICML.cc/2026/Conference — ICML 2026 regular_

### Official Review · Reviewer_23Wn · 2026-02-17

**Soundness:** 3
**Presentation:** 3
**Significance:** 3
**Originality:** 3
**Overall Recommendation:** 5
**Confidence:** 3

**Summary:**

This paper introduces a novel planning procedure, "Twice Sequential Monte Carlo Tree Search," (TSMCTS) which can be used in AlphaZero-like model-based reinforcement learning algorithms to replace MCTS. The method is based on the algorithm by Piché et al. (2019), which introduces the Sequential Monte Carlo Planning algorithm, an application of particle filters to the Control-As-Inference (CAI) framework. The TSMCTS algorithm is an extension of this idea that uses a more general greedification operator in place of the CAI concept of "optimality".

**Compliance With Llm Reviewing Policy:**

Affirmed.

**Final Justification:**

I think the paper is interesting and represents a significant contribution to the field. The rebuttal addressed my questions and reinforced my prior assessment. I thus recommend acceptance to ICML.

**Key Questions For Authors:**

My own background in model-based RL is limited, so I read up on MCTS and SMC methods before reading this submission in detail. The final algorithm (TSMCTS) is relatively complex, so the authors introduce intermediate algorithms (RL-SMC and SMCTS) as stepping stones. My questions concern the RL-SMC algorithm (Algorithm 1), since TSMCS is directly based on this algorithm.
1. Is there a use case for RL-SMC? In particular, it looks to me like the output of this algorithm is the policy $\pi_{SMC}^T$, which seems the be the policy that follows $\pi'$ for $T$ steps and thereafter follows $\pi_\theta$. Since it is necessary to compute $\pi'$ anyway (Line 4 of Algorithm 1), why not just use $\pi'$ directly? Am I misunderstanding something?
2. Line 6 of Algorithm 1 is a bit confusing. I suppose this is the "resampling" step, so what is sampled is not really the tuple $(j, a, s')$, but instead the existing tuples are simply relabeled. Is this correct?
3. Why are the $j$ variables initialized as $j_1^n = n$? And is the update for $j$ simply $j_{t + 1} = j_t$?
4. What is the purpose of the 'selection period $M$' in Algorithm 1? If I understand correctly, the standard "sequential importance resampling" (SIR) algorithm is the special case where $M = 1$?

**Limitations:**

yes

**Strengths And Weaknesses:**

Overall, the paper is well-written and the development of the algorithm is well-motivated. The method is original and interesting, combining previous techniques from MCTS, particle filtering, and bandit algorithms. The empirical results look very promising, and I appreciate that the authors included extensive comparisons. I also checked the proofs in appendix A, and they seem correct. I believe that the TSMCTS algorithm represents a significant contribution to the model-based RL toolkit, and based on this, I recommend acceptance to ICML. However, in my opinion, the presentation in the paper can (and should) be improved to enhance clarity. In particular, see the questions section below. I am thus giving a score of 4, and am willing to raise it.

---

> ### Author Rebuttal · Authors · 2026-03-30
>
> We thank the Reviewer for their useful clarifying questions and the time spent reviewing our work.
>
> > Is there a use case for RL-SMC? Is $\pi^T_{SMC}(a|s) \neq \pi'(a|s)$?
>
> Yes,  RL-SMC with $\pi^T_{SMC}(a|s_1) $ has a real use case for policy improvement over the one-step $\pi'(a|s_1) $ which is $ \neq \pi^T_{SMC}(a|s_1)$.
>
> This is because $\pi^T_{SMC}(a|s_1) $ is effectively $ \propto \pi_\theta(a|s_1) \exp( Q^{\pi'}(s_1,a))  \neq \pi'(a|s_1) \propto \pi_\theta(a|s) \exp( Q^{\pi_\theta}(s,a)) $ (note $Q^{\pi'}$ in the first expression, and $Q^{\pi_\theta}$ in the second).
>
> The reason for that is that the expression used to extract the policy improvement after timestep $T$ (Equation 7) is effectively an expectation under the target $p_T(\tau_T)$.
> We have: $ \pi^T_{SMC}(a|s) \approx \hat \pi^T_{SMC}(a | s) := \sum_{n=1}^N \bar w_{T}^{n}1_{\tau_T^n(a_1) = a} $, where the weights contain the returns of $\pi'$ on trajectories $\tau_T$ and are essentially: $w_T \propto \exp(Q^{\pi'}(s,a)) $.
>
> In fact, RL-SMC is exactly the algorithm used by [1,2].
>
> We will make sure this is made more clear in the paper.
>
> > Line 6 of Algorithm 1 is a bit confusing. I suppose this is the "resampling" step, so what is sampled is not really the tuple $(j, a, s')$, but instead the existing tuples are simply relabeled. Is this correct?
>
> This is indeed the *resampling step* [2] (we opted to use the terminology used by [3], "selection", but both terms are common for this operation, to our knowledge).
> This can indeed be thought of as effectively a relabeling step.
>
> However, this relabeling is happening by sampling with replacement from the Multinomial induced by $\bar w_t$.
> E.g., the particles are re-sampled from this Multinomial.
> As a result (an overall intended result), unlikely trajectories $\tau^n_t$ drop out and replaced by duplicates (which causes path degeneracy, but reduces the variance over steps $T$).
>
> > Why are the $j$ variables initialized as $j_1^n = n$? And is the update for $j$ simply $j_{t + 1} = j_t$?
>
> At the first step $t = 1$, every ancestor label $j_1^n$ is unique: each particle is associated with its own ancestor-trajectory. So we set $j_1^n = n $.
>
> $j_t$ is indeed only modified in the resampling step, from the Multinomial induced by $\bar w_t$.
> We will clarify in the pseudo code. Thanks for pointing that out.
>
> > What is the purpose of the selection period?
>
> The selection/resampling introduces variance in the current step, but significantly reduces aggregated variance over steps [4].
> Selection is also the direct cause of path degeneracy: the more selection steps, the more opportunity for degeneracy.
>
> The selection period allows tuning SMC to trade off between *reduced variance* and *reduced path degeneracy* and was used by prior work [1, 2, 5], which we follow to keep comparisons as direct as possible.
>
> References:
>
> [1] Macfarlane, Matthew V., et al. "SPO: Sequential Monte Carlo Policy Optimisation." NeurIPS (2024).
>
> [2] De Vries, Joery A., et al. "Trust-Region Twisted Policy Improvement." ICML (2025).
>
> [3] Chopin, Nicolas. "Central limit theorem for sequential Monte Carlo methods and its application to Bayesian inference." The Annals of Statistics (2004).
>
> [4] Chopin, Nicolas, and Omiros Papaspiliopoulos. An introduction to sequential Monte Carlo. Springer (2020).
>
> [5] Piché, Alexandre, et al. "Probabilistic planning with sequential monte carlo methods." ICLR (2018).

---

> > ### Author Rebuttal · Reviewer_23Wn · 2026-04-03
> >
> > I thank the authors for their detailed response and am happy to raise my score to accept.

---

### Official Review · Reviewer_43Da · 2026-03-13

**Soundness:** 3
**Presentation:** 3
**Significance:** 3
**Originality:** 3
**Overall Recommendation:** 4
**Confidence:** 4

**Summary:**

TSMCTS is a search algorithm for model-based RL that addresses high variance and path degeneracy in SMC-based tree search. It combines Sequential Halving (to reduce variance by searching over actions at each depth) with a "twice" SMC procedure (running independent searches per root action). Experiments across discrete and continuous environments show strong performance scaling with search depth, lower variance, and better policy targets compared to SMC baselines and GumbelMCTS.

**Compliance With Llm Reviewing Policy:**

Affirmed.

**Final Justification:**

The authors adequately addressed by concerns and I maintain that this is a nice paper and keep my score with increased confidence.

**Key Questions For Authors:**

See weaknesses

**Strengths And Weaknesses:**

Strengths
- The paper cleanly separates the variance and path degeneracy issues in SMC search, and proposes targeted solutions for each.
- TSMCTS outperforms or matches all baselines across all five environments (Figure 2), which is a strong result.
- Well-written and well-structured paper: The formulation of SMC for RL beyond CAI (Section 3-4) is clearly presented and the solution in Section 5 is also clear.

Weaknesses
- All experiments evaluate TSMCTS as a one-shot policy improvement operator given fixed policy and value networks. I would be curious to see results for using TSMCTS within a full AlphaZero-style training loop.
- The paper argues that TSMCTS retains SMC's parallelization advantages and has the same asymptotic complexity, but there is no actual GPU runtime data
- I found the continuous action space handling underspecified. How sensitive are we to the number of the candidate actions?
- It'd be nice to see results with imperfect learned dynamics models, which would better reflect the practical model-based RL setting.

---

> ### Author Rebuttal · Authors · 2026-03-30
>
> We thank the Reviewer for their questions and comments and the time spent reviewing our work.
>
> > Are the experiments limited to using TSMCTS as a one-shot policy improvement operator?
>
> Please note that **all the main experiments (Figure 1 left, Figure 2, 3 4 and 5) use TSMCTS within a full AlphaZero-style training loop** (line 373, Algorithm 4), and **are not** limited to using TSMCTS as a one-shot policy improvement operator.
>
> *Only* Figure 1 center and right, the ablations which evaluate variance reduction and path degeneracy mitigation, evaluate TSMCTS as a one-shot policy improvement operator given fixed policy and value networks.
>
> We will make sure this is made more clear in the paper.
>
> > GPU runtime:
>
> Runtime data on GPUs is presented in Figure 4.
>
> Runtime complexity is analyzed analytically in Appendix A.8 and shown to be the same as the SMC baseline.
>
> > Continuous action space handling - sensitivity to candidate actions?
>
> We have indeed not observed strong sensitivity on the number of candidate actions $m_1$ as long as $m_1 > 2$, as shown in Figure 3.
> The SMC baseline is shown to not be very sensitive to the number of candidate actions in continuous-action environments (Figure 3 left in [1]), except that it cannot scale with depth without sufficient particles.
> In contrast, TSMCTS scales gracefull with depth (Figure 1 left) and overall can be expected to be more stable than the SMC baseline due to the address of variance and path degeneracy.
>
> > It'd be nice to see results with imperfect learned dynamics models, which would better reflect the practical model-based RL setting:
>
> The original work which used SMC for search for action selection [2] used learned dynamics, while recent follow-up work [1,3] upon which TSMCTS is built and designed to enhance focused on true dynamics.
> Given the success with both learned and true dynamics demonstrated by SMC as well as other search algorithms [2,3,4,5,6] we would expect TSMCTS to work well with learned dynamics as well.
>
> However, since the main aim of this work is to solve problems in the planner used by [1,3] and allow it to scale with $T$, we focus here on the same experimental setup and domains used by [1,3].
>
> References:
>
> [1] Macfarlane, Matthew V., et al. "SPO: Sequential Monte Carlo Policy Optimisation." NeurIPS (2024).
>
> [2] Piché, Alexandre, et al. "Probabilistic planning with sequential monte carlo methods." ICLR (2018).
>
> [3] De Vries, Joery A., et al. "Trust-Region Twisted Policy Improvement." ICML (2025).
>
> [4] Silver et al. "A general reinforcement learning algorithm that masters chess, shogi, and Go through self-play." Science (2018).
>
> [5] Schrittwieser, Julian, et al. "Mastering atari, go, chess and shogi by planning with a learned model." Nature (2020).
>
> [6] Danihelka, Ivo, et al. "Policy improvement by planning with Gumbel." ICLR (2022).

---

> > ### Author Rebuttal · Reviewer_43Da · 2026-04-03
> >
> > Thanks for the responses. That's helpful and I continue to think that this is a nice contribution and keep my score.

---

### Official Review · Reviewer_gNvV · 2026-03-13

**Soundness:** 2
**Presentation:** 2
**Significance:** 2
**Originality:** 2
**Overall Recommendation:** 4
**Confidence:** 3

**Summary:**

This paper proposes TSMCTS (Twice Sequential Monte Carlo Tree Search), a search algorithm for policy improvement in model-based RL that addresses variance growth and path degeneracy in standard SMC-based search. The method has three layers: (1) RL-SMC (Section 3), which reformulates the SMC target/proposal outside Control-As-Inference using a general greedification operator; (2) SMCTS (Section 4), which maintains running-mean Q-values at the root via backpropagation, retaining value information even after particle collapse; and (3) TSMCTS (Section 5), which wraps SMCTS inside Sequential Halving at the root, progressively halving candidate actions while doubling the per-action particle budget. The paper provides asymptotic policy improvement guarantees (Theorems 1–2) and evaluates on discrete (Snake, Rubik's Cube) and continuous (Humanoid, Ant, HalfCheetah) environments using true dynamics models.

**Compliance With Llm Reviewing Policy:**

Affirmed.

**Final Justification:**

The authors addressed all my concerns and I raise my evaluation for this paper.

**Key Questions For Authors:**

- How sensitive are results to particle count $N$? The main experiments use $N = 4$; does the relative advantage of TSMCTS over baselines grow, shrink, or remain stable as $N$ increases to 16, 32, or 64?

- Can you explain further why you can make the claim about stationarity argument (Appendix A.5)  as we all know MCTS settings are non-stationary?

**Limitations:**

The authors acknowledge the limitation to true dynamics models but do not adequately discuss the implications. The restriction to simple environments and very low particle counts limits the generalizability of the conclusions.

**Strengths And Weaknesses:**

Strengthens:

The paper addresses real and well-documented problems - variance growth with search depth and path degeneracy - in SMC-based planning. The diagnosis is clear and well-motivated.
The SMCTS backpropagation idea is the most original contribution. By maintaining running-mean Q-values \bar{Q}_t(s_1, a) at the root, the algorithm naturally retains information about actions whose particles have been resampled away, simultaneously addressing path degeneracy and reducing variance. This is a simple but effective insight.

The observation that SMC provides stationary returns - unlike MCTS where returns are non-stationary due to tree growth - and therefore better matches Sequential Halving's theoretical assumptions (Appendix A.5) is valid and non-obvious. While SH at the root is borrowed from GumbelMCTS (Danihelka et al., 2022), the justification for why it fits SMC better than MCTS is a genuine contribution.
The experimental design is commendable: all agents use the same DNN architecture, hyperparameters, and true dynamics model, isolating the search procedure as the only variable. The scaling-with-depth experiments (Figure 1, left) are the paper's strongest empirical result, clearly showing that baseline SMC degrades while TSMCTS scales gracefully.

Weaknesses

- The individual algorithmic components are largely drawn from prior work. RL-SMC (Section 3) is primarily a notational simplification - the weight update $w_t^n \propto \pi'(a_t^n|s_t^n)/\pi_\theta(a_t^n|s_t^n)$ is standard importance sampling, and the authors themselves show it reduces to CAI-SMC in Appendix A.2. The SMCTS backpropagation idea is conceptually incremental over TRT-SMC (de Vries et al., 2025), which already retains value information for root actions after path degeneracy by saving the last return per action; SMCTS improves this by averaging all returns, but the conceptual step is small. Sequential Halving at the root is directly borrowed from GumbelMCTS with the stationarity argument (Appendix A.5) being the main new observation - interesting but relatively minor. The question is whether the synergy of these known components constitutes a sufficient contribution.

- Limited experimental scope: All experiments use only 4 particles in the main comparison (Figure 2); scaling behavior with $N = 16, 32, 64$ is not investigated. or maybe I miss it.

- The comparison against GumbelMCTS may not be fully competitive: MCTS builds a tree that grows over iterations with information reuse, while SMC uses independent trajectories, making equating the two purely by forward-pass count not fair The main motivation for SMC over MCTS is GPU-friendliness, yet the runtime comparison (Figure 4) provides only wall-clock times on unspecified hardware with no GPU utilization metrics, throughput comparisons, or scaling with batch size. The parallelization advantage is mostly claimed rather than demonstrated.

---

> ### Author Rebuttal · Authors · 2026-03-30
>
> We thank the Reviewer for their comments and questions and the time spent reviewing our work.
>
> > Novelty:
>
> A significant part of our contribution (and its novelty) is in improved perspective:
>
> 1. Our novel derivation of RL SMC is the first to show (to our knowledge) that SMC can be used without CAI and its restrictions, with any greedification operator, enabling the interpretation of SMC as a general policy improvement operator.
> 2. This perspective is supported by novel theoretical results (Thm 1 and Cor 1).
> 3. Supported by RL-SMC, we switch the perspective of the search from a distribution over trajectories to values at the root, which facilitates incorporating the mechanisms for variance reduction:
> 4. Aggregating the values within SMC (which as noted is novel).
> 5. Aggregating values with SH across iterations of SMC (which as noted is novel), providing dramatic performance improvement (Figure 1 left) and allowing SMC to scale with sequential compute, an important result to publish.
> 6. Additionally, SH with particles is itself a novel algorithm, irrespective of SMC, to our knowledge.
> 7. Theorem 2 is novel and provides theoretical grounding for popular modern algorithms such as Gumbel/Alpha/MuZero [1,5] which improve the policy at the root using greedification with respect to the value of a different, improved policy.
>
> > Low $N$, sensitivity to and scaling with $N > 4$:
>
> Please note that **Figures 1 and 3, which investigate scaling, aggregate across $N=4,8,16$** showing that TSMCTS scales gracefully with $T$ irrespective of and not sensitive to $N=4$. Unlike the baseline, which strongly depends on $N$ and more so as depth grows (Figure 3 in [4]).
>
> When comparing to non-SMC baselines (Figure 2), we followed [2] which used $N=4$ to keep the comparison as direct as possible:
> As $N$ increases, the comparison to MCTS becomes less direct: MCTS iterates $NT$ sequentially, while SMC iterates $T$ sequentially and $N$ in parallel.
> Similarly model free RL cannot be scaled with increased compute in the same manner as increasing $N$.
>
> > Comparison to GumbelMCTS may not be fully fair:
>
> We agree: in our understanding it *favors MCTS*.
> It would be fairer to allow MCTS $T$ iterations, the same as SMC variants [2,3,4], instead of $NT$, which presumes MCTS can iterate $N$ in parallel.
> Nonetheless, we followed [2, 4] which gave MCTS this advantage because it strengthens the significance of TSMCTS outperforming MCTS.
> We show empirically that this comparison indeed unfairly favors MCTS which induces a larger runtime for the same budget, in Figure 4.
>
> > Runtime comparison on GPUs:
>
> All seeds in Figure 4 used GPUs from the same set (A40s, for the most part) and the results are aggregated across seeds, 20 seeds per agent, where TMCTS compares to SMC with modest overhead and is significantly more runtime efficient than MCTS.
> All implementations (TSMCTS, SMC, MCTS) use GPU acceleration with Jax (line 411, right column). The implementation was optimized by [2] to compare SMC and MCTS GPU runtime, and was not modified by us except for the addition of TSMCTS (line 378, left column).
>
> [4] investigated the scaling of SMC for search with parallel compute and the respective comparison to MCTS more extensively.
>
> In Appendix A.8 we demonstrate analytically that TSMCTS has the same runtime / space complexity as SMC and better runtime / space complexity than MCTS.
>
> > Stationarity:
>
> In TSMCTS $Q^{\pi^T_{SMCTS}}(s,a)$ the mean of each "arm" $a$ is the same at every iteration $i$ (lines 323-328, e.g. stationary).
> This is because SMC/TSMCTS does not save the search tree in memory and thus does not modify the search policy between iterations.
> In contrast in MCTS the mean $Q^{\pi_i}(s,a)$ changes every iteration $i$.
>
> > Implications of true dynamics:
>
> Due to the success of SMC with learned models [3] as well as true models [2,4], and as TSMCTS relies on mechanisms that work well in learned and true dynamics [5], we do not believe there are significant implications to evaluating TSMCTS on true dynamics.
>
> > The restriction to simple environments limits generalizability.
>
> We chose to evaluate TSMCTS specifically in the same domains for which the baselines where originally designed and tuned to strengthen the statistical significance of the results.
> To support generalizability, the domains chosen by prior work are varied and include discrete rich and sparse reward (Snake and Rubikscube respectively) as well as popular classical continuous control (humanoid, ant, halfcheetah).
>
> References:
>
> [1] Grill, et al. "Monte-carlo tree search as regularized policy optimization." ICML (2020).
>
> [2] De Vries, et al. "Trust-Region Twisted Policy Improvement." ICML (2025).
>
> [3] Piché, et al. "Probabilistic planning with sequential monte carlo methods." ICLR (2018).
>
> [4] Macfarlane, et al. "SPO: Sequential Monte Carlo Policy Optimisation." NeurIPS (2024).
>
> [5] Danihelka, et al. "Policy improvement by planning with Gumbel." ICLR (2022).

---

> > ### Author Rebuttal · Reviewer_gNvV · 2026-04-03
> >
> > I thank the authors for addressing all my concerns and I will raise my evaluation.

---

### Official Review · Reviewer_MWty · 2026-03-15

**Soundness:** 3
**Presentation:** 3
**Significance:** 3
**Originality:** 2
**Overall Recommendation:** 4
**Confidence:** 3

**Summary:**

The paper introduced a sequential monte carlo (SMC) search method for policy improvement in model-based RL. It addresses two limitations  of existing SMC methods: large variance and path degeneracy. The variance is reduced by using average Q values during search and the path degeneracy is mitigated by incorporating an existing method, Sequential Halving (SH). The resulting algorithm is  named Twice Sequential Monte Carlo Tree Search (TSMCTS).
Experiments in discrete action and continuous environments  show that the proposed method outperforms baselines in most environments, scales better with compute, while retaining properties of SMC such as easy to parallelize.

**Compliance With Llm Reviewing Policy:**

Affirmed.

**Final Justification:**

I thank the authors for the further comments. The rebuttal has improved my understanding of this paper and I think that it brings decent contribution hence my positive score. The reason for not scoring higher is that in my opinion, the attribution of the performance increase to variance reduction and reduced path degeneracy is not supported by sufficient empirical evidence.

**Key Questions For Authors:**

1. Can you provide details on how the hyperparameters were chosen for $m_1$ and $\beta_{\text{root}}$?
2. According to Table 6, the hyperparameter $m_1$ is set to 16 for Fig. 1 Left, but to 4 for all others. What’s the reason for such a difference?
3. What're the limitations of the propose method?

**Limitations:**

The authors haven't discussed the limitations of the proposed method. It would be helpful to explicitly state limitations and analyze the observed failure cases if any.

**Strengths And Weaknesses:**

**Strengths:**
1. Well written, easy to follow and clearly motivated
2. Better empirical performance with a small increase in computational overhead


**Weaknesses:**

The main area for improvement is the experiments and the analysis of the results.
1. The claimed benefit of the proposed algorithm is (1) reduced variance of root estimator, (2). reduced path degeneracy. However, the direct empirical evidence supporting the claim is only from one single environment Snake, in Figure. 1 (mid, right).
2. Line 437 states that “TSMCTS-based agents significantly outperform or match all baselines”.  It appears to be over-stating because it’s not matching in Rubikscube, and also slightly underperforms in halfcheetah.
3. There also seems to be limited analysis or discussion about why the proposed method outperforms greatly for some environments but not the others, and how that difference might relate to the variance and path degeneracy..

Discussion about related works

4. I’m a bit confused about the discussion of TD-$\lambda$ in the related works section. In particular, the popularity of TD-$\lambda$ isn’t necessarily related to whether it is critical to the algorithm performance. It would be better to justify it or cite prior works if that had been previously discussed in the literature.

---

> ### Author Rebuttal · Authors · 2026-03-30
>
> We thank the Reviewer for their comments and the time spent reviewing the presented work.
>
> > Empirical evaluation of variance and path degeneracy in Snake.
>
> We chose to run the experiment in Snake as the most informative environment: rich reward (unlike single-goal Rubikscube) and discrete action.
> The benefits of TSMCTS are shown quantitatively across all environments in Figures 1 left and 2.
>
> We include analytical motivation for variance reduction and path degeneracy mitigation in Appendices A.6 & A.7.
>
> > Matching or outperforming baselines.
>
> In Cheetah the confidence bounds overlap, e.g. no statistically significant difference.
> In cube the final performance overlaps, which we view as comparable.
> We will soften this statement into "In all environments TSMCTS-based agents significantly outperform or perform comparably to all baselines." Does this address the concern?
>
> > Limited analysis why the method outperforms more in some environments over others.
>
> This remains a difficult, open question in RL for the majority of algorithms, presumably due to the large number of independent effects unique to each environment. It is the main reason popular modern [1,2,3,4,5,6,7,8] and classical [9,10,11] methods evaluate in many domains and do not include analysis of the reasons behind the specific performance in each domain.
>
> > The discussion of TD-$\lambda$ is confusing.
>
> We thank the reviewer for pointing that out.
> We will clarify in the paper: TD-$\lambda$ is a reasonable direction for future work in TSMCTS for further variance reduction, which we chose not to pursue due to [12].
> [12] shows that in MCTS, which uses the same value aggregation as SMCTS, TD-$\lambda$ does not have a significant performance contribution.
>
> > Choosing $m_1$ and $\beta_{root}$.
>
> $m_1$ is searched over in Figure 3.
> We conclude that it's necessary for $m_1 \geq 4$ (line 403, right column).
> Note that $m_1 \leq N$ because particle-based SH cannot search more actions in parallel than there are particles, so $m_1 \gets \min(N, m_1) $ is set.
>
> $\beta_{root}$ was chosen using a rough grid search (line 1148). The effect of $\beta_{root}$ is akin to a learning rate: the larger, the greedier the policy and the larger the difference between the prediction $\pi_\theta$ and the target $\pi_{search}$. The smaller, the more conservative the update.
>
> > Why was $m_1 = 16$ chosen for Figure 1 and $m_1 = 4$ for the rest?
>
> $m_1 = 16$ showed slightly-better scaling with depth (Figure 3).
> We've chosen it to demonstrate the scaling potential of the algorithm: as $N$ increases, practitioners should consider increasing $m_1$.
> For the other figures we've chosen $4$ for consistency, as it allows the agents to use the same hyperparameter.
> Otherwise, each budget $N$ is evaluated with different $m_1 \gets \min(N,16) = N $, rather than the same $m_1=4$.
>
> > It would be helpful to explicitly state limitations and analyze the observed failure cases if any.
>
> To the best of our understanding, our method does not introduce significant limitations to the SMC line of work [6,7,13].
> The runtime complexity of TSMCTS is the same as the baseline SMC (appendix A.8).
> Empirically, the runtime of TMSCTS introduces a minor overhead (Figure 4).
> Analytically, RL-SMC simplifies the development and motivation of SMC for search compared to the presentation and theoretical motivation of CAI used by [6,7,13].
> An argument could be made as to a slight increase in implementation complexity, with the introduction of SH.
> Similar to prior work [5], we did not view this minor challenge as sufficiently relevant to discuss.
>
> References:
>
> [1] Hafner, et al. "Mastering diverse control tasks through world models." Nature (2025).
>
> [2] Fujimoto, et al. "For sale: State-action representation learning for deep reinforcement learning." NeurIPS (2023).
>
> [3] Hansen, et al. "TD-MPC2: Scalable, robust world models for continuous control." ICLR (2024).
>
> [4] Wang, et al. "Bootstrapped Model Predictive Control." ICLR (2025).
>
> [5] Danihelka, et al. "Policy improvement by planning with Gumbel." ICLR (2022).
>
> [6] Macfarlane,et al. "SPO: Sequential Monte Carlo Policy Optimisation." NeurIPS (2024).
>
> [7] De Vries, et al. "Trust-Region Twisted Policy Improvement." ICML (2025).
>
> [8] Wang, et al. "EfficientZero V2: Mastering Discrete and Continuous Control with Limited Data." ICML (2024).
>
> [9] Fujimoto, et al. "Addressing function approximation error in actor-critic methods." ICML (2018).
>
> [10] Haarnoja, et al. "Soft actor-critic: Off-policy maximum entropy deep reinforcement learning with a stochastic actor." ICML (2018).
>
> [11] Abdolmaleki, et al. "Maximum a Posteriori Policy Optimisation." ICLR (2018).
>
> [12] Khandelwal, et al. "On the analysis of complex backup strategies in monte carlo tree search." ICML (2016).
>
> [13] Piché, et al. "Probabilistic planning with sequential monte carlo methods." ICLR (2018).

---

> > ### Author Rebuttal · Reviewer_MWty · 2026-04-04
> >
> > I thank the authors for the rebuttal. It has addressed most of my concerns. The authors pointed to prior works that do not analyze environment-specific performance or discuss certain limitations. While I appreciate that context, I do not think it justifies omitting such discussion here or addresses the underlying concern. Overall, the rebuttal was sufficient for me to maintain my original score.

---

> > > ### Author Response · Authors · 2026-04-04
> > >
> > > We thank the Reviewer for acknoweldging our rebuttal.
> > >
> > > > The authors pointed to prior works that do not analyze environment-specific performance or discuss certain limitations. While I appreciate that context, I do not think it justifies omitting such discussion here or addresses the underlying concern.
> > >
> > > **In regards to limitations of TSMCTS compared to the SMC baseline and differences in performance between TSMCTS and baseline SMC in different environments:**
> > >
> > > TSMCTS significantly outperforms the baseline SMC on which it was designed to improve (in sample efficiency and final-observed performance), in all environments (bar ant and humanoid, where it outperforms significantly only in sample efficiency), in an all-else-equal, apples-to-apples comparison (Figure 5).
> > >
> > > This, without introducing any limitations we are able to point to, such as additional runtime complexity, space complexity, bias, etc.
> > >
> > > *We conclude that **variance reduction** and **path degeneracy address** are important and have a significant effect across environments in this experiment suite.*
> > >
> > > We thank the reviewer for emphasizing this point, and will add this observation to the paper.
> > >
> > > **In regards to the limitations of SMC-based algorithms in general and differences in performance in different environments compared to other baselines:**
> > >
> > > The closest point of comparison is MCTS.
> > > Ultimately, the relationship between MCTS-based planners that retain the search tree and more light weight and parallelizable planners such as SMC is complex, since the algorithms maintain differences whose effects are hard to predict:
> > >
> > > 1. As MCTS retains the search tree, it is able to iteratively greedify the policy inside the tree, while SMC is more computationally lightweight, does not retain the search tree and can only modify the search policy once at each node.
> > >
> > > 2. MCTS dynamically allocates its search budget one way (iterative re-search from the root with a non-stationary policy), while SMC dynamically allocates in a different way (resampling periodically proportional to particle weights).
> > >
> > > 3. SMC is forced to explore during search in a certain way (all particles are sampled in parallel from the proposal) and average across all trajectories proportional to their probabilities under the *improved* policy. MCTS always propagates all returns that it has seen with equal weights, and balances exploration and exploitation in a different manner (the search policy is explicitly designed for a specific exploration-exploitation tradeoff).
> > >
> > > 4. MCTS dedicates a dynamic depth depending on the search policy and the rate at which it concentrates with iterations while SMC algorithms have a constant depth.
> > >
> > > The first work (to our knowledge) to compare SMC with MCTS-based AlphaZero, [6], hypothesized that the major differences in performance of MCTS itself across different environments was possibly a result of MCTS struggling to generalize across domains when using the same hyperparameters in all domains (the experimental setup used for the comparison).
> > >
> > > As the purpose of our work is to design better SMC planners for policy improvement, identifying the specific contrasts in performance between TSMCTS and baselines outside of SMC goes beyond the scope of this work.
> > >
> > > References (numbering retained from original rebuttal for clarity):
> > >
> > > [6] Macfarlane,et al. "SPO: Sequential Monte Carlo Policy Optimisation." NeurIPS (2024).

---

### Decision · Program_Chairs · 2026-04-30

**Decision:**

Accept (regular)

**Comment:**

This paper proposes a variant of Monte Carlo Tree Search based on a twice sequential Monte Carlo (TSMCTS) framework, aiming to address variance and path degeneracy issues in planning and decision-making. The reviewers generally acknowledge the technical soundness and the novelty of introducing a principled SMC-based formulation into tree search, with empirical results showing consistent improvements over several baselines across multiple environments. In particular, the method appears to provide clearer advantages in sparse-reward and discrete-action settings, and the rebuttal clarifies key design choices such as variance reduction and degeneracy mitigation.

Some concerns were raised regarding the limited scope of experimental domains and the added complexity of the framework (e.g., additional hyperparameters and coupling between components), as well as whether the empirical gains are uniformly significant across all settings. However, the authors addressed several of these concerns by clarifying comparisons, tempering claims, and providing additional explanations in the rebuttal. While the method may not yet demonstrate universal superiority, it offers a meaningful and well-motivated extension to existing MCTS approaches.

Overall, the paper presents a technically solid and conceptually interesting contribution that bridges SMC methods and tree search in a novel way. Given its potential to inspire further research and its demonstrated empirical benefits in key scenarios, I recommend acceptance.